# EA-HAS-Bench: Energy-aware Hyperparameter and Architecture Search Benchmark

**Shuguang Dou**[1] [*] **Xinyang Jiang**[2] [†] **Cairong Zhao**[1] [‡] **Dongsheng Li**[2]
[1] Tongji University, [2] Microsoft Research Asia

## Abstract

The energy consumption for training deep learning models is increasing at an alarming rate due to the growth of training data and model scale, resulting in a negative impact on carbon neutrality. Energy consumption is an especially pressing issue for AutoML algorithms because it usually requires repeatedly training large numbers of computationally intensive deep models to search for optimal configurations. This paper takes one of the most essential steps in developing energy-aware (EA) NAS methods, by providing a benchmark that makes EA-NAS research more reproducible and accessible. Specifically, we present the first large-scale energy-aware benchmark that allows studying AutoML methods to achieve better trade-offs between performance and search energy consumption, named EA-HAS-Bench. EA-HAS-Bench provides a large-scale architecture/hyperparameter joint search space, covering diversified configurations related to energy consumption. Furthermore, we propose a novel surrogate model specially designed for large joint search space, which proposes a Bézier curve-based model to predict learning curves with unlimited shape and length. Based on the proposed dataset, we modify existing AutoML algorithms to consider the search energy consumption, and our experiments show that the modified energy-aware AutoML methods achieve a better trade-off between energy consumption and model performance.

## 1 Introduction

As deep learning technology progresses rapidly, its alarming increased rate of energy consumption causes growing concerns (Schwartz et al., 2020; Li et al., 2021a; Strubell et al., 2019). Neural architecture search (NAS) (Elsken et al., 2019), hyperparameter optimization (HPO) (Feurer & Hutter, 2019) lifted the manual effort of neural architecture and hyperparameter tuning but require repeatedly training large numbers of computationally intensive deep models, leading to significant energy consumption and carbon emissions. For instance, training 10K models on CIFAR-10 (Krizhevsky et al., 2009) for 100 epochs consume about 500,000 kWh of energy power, which is equivalent to the annual electricity consumption of about 600 households in China.

As a result, it is essential to develop search energy cost aware (EA) AutoML methods, which are able to find models with good performance while minimizing the overall energy consumption throughout the search process. However, existing NAS studies mainly focus on the resource cost of the searched deep model, such as parameter size, the number of float-point operations (FLOPS), or latency (Tan et al., 2019; Wu et al., 2019; He et al., 2021). Exploiting the trade-off between model performance and energy cost during the searching process has been rarely studied (Elsken et al., 2019). In this paper, we propose to take one of the most essential steps in developing energy-aware (EA) NAS methods that make EA-NAS research more reproducible and accessible. Specifically, we provide a benchmark for EA-NAS called Energy Aware Hyperparameter and Architecture Search Benchmark (EA-HAS-Bench), where the researchers can easily obtain the training energy cost and model performance of a certain architecture and hyperparameter configuration, without actually training the model. In order to support developing energy-aware HPO and NAS methods, the proposed EA-HAS-Bench should satisfy three requirements.

---

[*]Work done during an internship in Microsoft Research Asia. Email: dousg@tongji.edu.cn.
[†]Corresponding authors. Email: zhaocairong@tongji.edu.cn, xinyangjiang@microsoft.com

Table 1: Overview of NAS benchmarks related to EA-HAS-Bench

| | Benchmark | Size | Bench. Type | Arch. Type | LCs | Query |
|---|---|---|---|---|---|---|
| NAS | NAS-Bench-101 | 423k | Tab. | Cell | | Accuracy |
| | NAS-Bench-201 | 6k | Tab. | Cell | ✓ | Accuracy & Loss |
| | NAS-Bench-301 | $10^{18}$ | Surr. | Cell | | Accuracy |
| | NATS-Bench | 32K | Tab. | Macro | ✓ | Accuracy & Loss |
| | HW-NAS-Bench | 15K/$10^{21}$ | Tab./Esti. | Cell | | Latency & Inference Energy |
| | | 423k | Surr. | Cell | ✓ | Accuracy |
| | NAS-Bench-x11 | 6k | Surr. | Cell | ✓ | Accuracy & Loss |
| | | $10^{18}$ | Surr. | Cell | ✓ | Accuracy |
| NAS+ HPO | NAS-HPO-Bench | 62K | Tab. | MLP | ✓ | Accuracy |
| | NAS-HPO-Bench-II | 192K | Surr. | Cell | | Accuracy |
| | EA-HAS-Bench (Ours) | $3 \times 10^{10}$ | Surr. | Macro | ✓ | Accuracy & Inference and Total Search Energy |

**Search Energy Cost** Our dataset needs to provide the total search energy cost of running a specific AutoML method. This can be obtained by measuring the energy cost of each particular configuration the method traverses and summing them up. As shown in Table 1, most of the existing conventional benchmarks (Ying et al., 2019; Dong & Yang, 2020; Siems et al., 2020) like NAS-Bench-101 do not directly provide training energy cost but use model training time as the training resource budget, which as verified by our experiments, is an inaccurate estimation of energy cost. HW-NAS-bench (Li et al., 2021b) provides the inference latency and inference energy consumption of different model architectures but also does not provide the search energy cost.

**Energy Related Joint Search Space** The search space of EA-HAS-Bench should include the configurations that affect both the model training energy cost and performance. Since both model architectures (e.g., scales, width, depth) and training hyperparameters (e.g., number of epochs) are correlated to the training energy cost, designing a joint hyperparameter and architecture search (HAS) search space is essential. Most NAS benchmarks use a single fixed training hyperparameter configuration for all architectures. Existing HAS benchmarks (Klein & Hutter, 2019; Hirose et al., 2021) use small toy search space which does not cover enough critical factors affecting the search energy cost. As a result, EA-HAS-Bench provides the first dataset with a ten-billion-level architecture/hyperparameter joint space, covering diverse types of configurations related to search energy cost.

**Surrogate Model for Joint Search Space** Due to the enormous size of the proposed joint search space, a surrogate model is needed to fill out the entire search space with a subset of sampled configurations. Existing surrogate methods (Zela et al., 2022; Yan et al., 2021) are not applicable to our proposed large-scale joint space. For instance, those methods do not consider the situation of maximum epoch variation and predict only a fixed-length learning curve or final performance. Thus, we propose the Bézier Curve-based Surrogate (BCS) model to fit accuracy learning curves of unlimited shape and length.

We summarize the contribution of this paper as follows:

- EA-HAS-Bench is the first HAS dataset aware of the overall search energy cost. [*]. Based on this dataset, we further propose a *energy-aware AutoML* method with search energy cost related penalties, showing energy-aware AutoML achieves a better trade-off between model performance and search energy cost.

- We provide the first large-scale benchmark containing an architecture/hyperparameter joint search space with over 10 billion configurations, covering various configurations related to search energy cost.

- We develop a novel surrogate model suitable for more general and complex joint HAS search space, which outperforms NAS-Bench-X11 and other parametric learning curve-based methods.

---

[*]The dataset and codebase of EA-HAS-Bench are available at https://github.com/microsoft/EA-HAS-Bench.

Table 2: Overview of EA-HAS-Bench's search space

| Type | Hyperparameter | Range | Quantize | Space |
|---|---|---|---|---|
| RegNet | Depth $d$ | [6,15] | 1 | 10 |
| | $w_0$ | [48, 128] | 8 | 10 |
| | $w_a$ | [8,32] | 0.1 | 241 |
| | $w_m$ | [2.5, 3] | 0.001 | 501 |
| | Group Width | [1, 32] | 8 | 5 |
| Total of Network Architectures | | | | $\approx 6 \times 10^7$ |
| Optima | Learning rate | {0.001, 0.003, 0.005, 0.01, 0.03, 0.05, 0.1, 0.3, 0.5, 1.0} | - | 10 |
| | Max epoch | {50, 100, 200} | - | 3 |
| | Decay policy | {'cos', 'exp', 'lin'} | - | 3 |
| | Optimizer | {'sgd', 'adam','adamw'} | - | 3 |
| Training | Data augmentation | {None,Cutout} | - | 2 |
| Total of Hyperparameter Space | | | | 540 |

## 2 CREATING ENERGY AWARE BENCHMARKS

### 2.1 EA-HAS-BENCH SEARCH SPACE

Unlike the search space of existing mainstream NAS-Bench that focuses only on network architectures, our EA-HAS-Bench consists of a combination of two parts: the network architecture space-RegNet (Radosavovic et al., 2020) and the hyperparameter space for optimization and training, in order to cover diversified configurations that affect both performance and energy consumption. The details of the search space are shown in Table 2.

**RegNet Search Space.** Our benchmark applies RegNet as the architecture search space because it contains several essential factors that control the scale and shape of the deep model, which corresponds to the training energy cost. Specifically, RegNet is a macro search space with the variation in width and depth explained by a quantized linear function. Specifically, for widths of residual blocks: $u_j = w_0 + w_a \times j$, where $0 \leq j \leq d$ and $d$ denotes the depth. An additional parameter $w_m$ is introduced to quantize $u_j$, i.e. $u_j = w_0 \times w_m^{s_j}$ and the quantized per-block widths $w_j$ is computed by

$$w_j = w_0 \times w_m^{\lfloor s_j \rceil} \tag{1}$$

where $\lfloor s_j \rceil$ denotes round $s_j$. the original search space of RegNet is for ImageNet and is non-trivial to directly apply to other datasets. As a result, we shrink down the original model size range of the RegNet space and constraint the total parameters and FLOPs of a model to a relatively small size, which achieves faster training and saves resources.

**Training hyperparameter Search Space.** Hyperparameter space (e.g., learning rate and maximum epoch) also has a great impact on energy consumption throughout the training phase. For example, the maximum epoch is almost proportional to training time, which is also proportional to training energy cost. Different learning rate also causes different convergence rate and different total training time, resulting in different training energy cost. Thus, for hyperparameter search space, EA-HAS-Bench considers training epochs, and the most important factors in training schemes, i.e., base learning rate, decay policy of learning rate, optimizer, and data augmentation.

As a result, the search space of EA-HAS-Bench contains a total of $3.26 \times 10^{10}$ configurations, including $6 \times 10^7$ architectures and 540 training hyperparameter settings with variant training epochs.

### 2.2 EVALUATION METRICS

The EA-HAS-Bench dataset provides the following three types of metrics to evaluate different configurations.

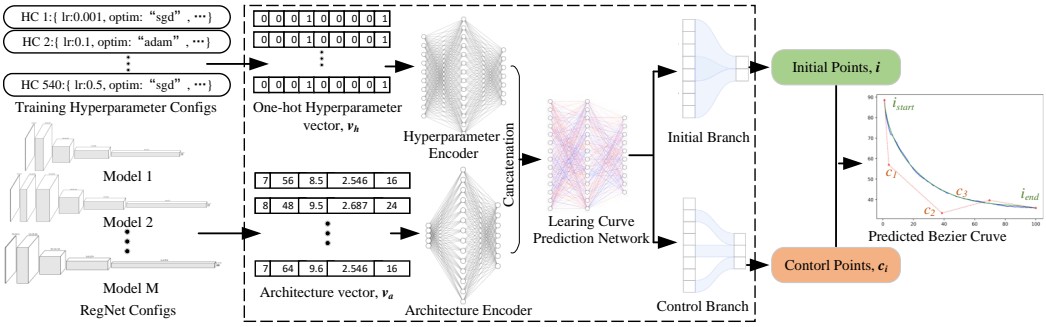

Figure 1: Overview of Bézier Curve-based Surrogate Model. HC denotes Hyperparameter configuration.

**Model Complexity.** Metrics related to model complexity include parameter size, FLOPs, number of network activations (the size of the output tensors of each convolutional layer), as well as the inference energy cost of the trained model.

**Model Performance.** In order to support multi-fidelity NAS algorithms such as Hyperband (Li et al., 2017), EA-HAS-Bench provides the full training information including training, validation, and test accuracy learning curves for each hyperparameter and architecture configuration.

**Search Cost.** Firstly, the energy cost (in kWh) and time (in seconds) to train a model under a certain configuration for one epoch are obtained. Then, by accumulating the energy consumption and runtime at each epoch, we obtain the total search cost of a configuration, which allows NAS/HPO methods to search optimal models under a limited cost budget.

## 2.3 BÉZIER CURVE-BASED SURROGATE MODEL

Due to the large EA-HAS-Bench search space, directly training all configurations in the space is infeasible even for a small dataset like CIFAR10. As a result, some of the metrics can not be directly measured, including the model performance curve, search energy cost, and runtime. Thus, similar to other recent works (Zela et al., 2022; Yan et al., 2021), we develop a surrogate model that expands the entire search space from a sampled subset of configurations.

As for energy cost and training time, we follow the Surrogate NAS Benchmark (Zela et al., 2022) and train LGB (LightGBM) (Ke et al., 2017) models to predict these. However, for learning curve prediction, surrogate models proposed by the existing NAS-Bench are not applicable to EA-HAS-Bench. Since EA-HAS-Bench contains various maximum training epochs in the search space, non of the existing surrogate model can cope with dimensionally varying inputs. More importantly, it is not possible to directly constrain the learning curve after using the dimensionality reduction operation (e.g., the error rate should be between 0 and 1). In our early experiments, the implementation of the NAS algorithm on the NAS-Bench-X11 surrogate model would yield results with more than 100% accuracy. Therefore, we propose a Bézier Curved-based Surrogate (BCS) Model to solve the above problems.

**Convert Learning Curve to Bézier Curve Control Points.** For each configuration of network architecture and training hyperparameters, the surrogate model outputs the learning curve containing the accuracy or error rate of each epoch. Inspired by the success of the Bézier curve in other areas (Liu et al., 2020), we choose the Bézier curve to fit learning curves of arbitrary length. The shape of a Bézier curve is entirely determined by its control points, and degree $n$ control points correspond to a Bézier curve of order $n-1$. The Bézier curve can be formulated in a recursive way as follows:

$$P(t) = \sum_{i=0}^{n} P_i B_{i,n}(t), t \in [0, 1] \tag{2}$$

where $P_i$ denotes control points, $B_{i,n}(t) = C_n^i t^i (1-t)^{n-i}$ and $i = 0, 1, \cdots, n$.

As a result, the task of regressing a learning curve of arbitrary length is simplified to predicting Bézier curve control points. Given a learning curve, $\{e_{x_i}, e_{y_i}\}_{i=1}^m$ where $e_y$ is the error of the $e_x$th epoch and $m$ is the maximum epoch, we need to get the optimal control points to generate Quartic Bézier curves to fit the learning curve. The control points are learned with the standard least square method. Since the horizontal coordinates of the start and end points of the learning curve are fixed (i.e., $i_{x_{start}} = 1$ and $i_{x_{end}} = $ maximum epoch), the surrogate model only predicts the vertical coordinates of these two control points. An illustration of generated Bézier curves is shown in Figure 7 in Appendix B.4.

**Surrogate Model Structure.** Given a RegNet architecture and hyperparameter configurations, BCS estimates the Bézier curve control points with a neural network. As shown in Figure 1, the proposed Bézier curve-based Surrogate model is composed of a hyperparameter encoder $E_h$, architecture encoder $E_a$, and learning curve prediction network $f$. The training hyperparameter configurations are represented as one-hot vector $v_h$ and then fed into $E_h$. The RegNet configuration parameters are normalized to values between 0 and 1, concatenated to a vector, and fed into $E_a$. Finally, the hyperparameter representation and architecture representation are fed into the learning curve predictor to estimate Bézier curve starting/ending points and control points:

$$\{i_{y_{start}}, i_{y_{end}}, c_{x_1}, c_{y_1}, \cdots, c_{x_3}, c_{y_3}\} = f(E_a(v_a), E_h(v_h)) \tag{3}$$

The learning curve predictor consists of two branches. One predicts the vertical coordinates of the starting and ending points of the Bézier curve, and the other branch predicts the other control points.

With the control points obtained, we can generate Bézier curves with equation (2), and then obtain the accuracy of each epoch based on the horizontal coordinates $e_{x_i}$ of the curve.

## 2.4 DATASET COLLECTION

Some previous works (Eggensperger et al., 2015) propose to sample more training data from the high-performance regions of the search space because an effective optimizer spends most of its time in high-performance regions of the search space. However, this sampling strategy causes a distribution shift between the sampled training data and the actual search space, which hurts the prediction accuracy of the surrogate model. As discussed and verified in recent work (Zela et al., 2022), a search space containing hyper-parameters is more likely to produce dysfunctional models which are rarely covered in a sampled sub-set focusing on high-performance regions, and hence purely random sampled training data yields more satisfactory performance. In summary, for EA-HAS-Bench's search space that contains both model architectures and hyperparameters, we use random search (RS) to sample unbiased data to build a robust surrogate benchmark.

The sampled architecture and hyperparameter configurations are trained and evaluated on two of the most popular image classification datasets, namely CIFAR-10 (Krizhevsky et al., 2009) and MicroImageNet challenge's (Tiny ImageNet) dataset (Le & Yang, 2015).

## 2.5 SURROGATE BENCHMARK EVALUATION

**Comparison Methods.** We compare the proposed method with six parametric learning curve based models (Domhan et al., 2015) ($Exp_3$, $ilog_2$, $pow_2$, log power, log linear, vapor pressure) and three surrogate models (SVD-XGB, SVD-LGB, SVD-MLP) from NAS-Bench-X11 Yan et al. (2021). For a fair comparison, the parametric learning curve-based model applies the same network structure as our proposed BCS. For NAS-Bench-X11, we compress learning curves of different lengths (50, 100, and 200 epochs) into the hidden space with the same dimension with three different SVDs respectively (although this is not convenient to cope with learning curves of arbitrary length). Tree-of-Parzen-Estimators (TPE) (Bergstra et al., 2011) is adopted for all surrogate models to search for the best hyperparameter configuration. The details of the experiments and ablation study are in Appendix B.

**Testing Set and Ground Truth (1 seed).** All surrogate model methods are evaluated on a separate testing set trained on two sets of random seeds. One of the two seeds of the test set is used as the ground truth, and data from the other seed can be seen as a tabular benchmark baseline (results in

Table 3: Compare Bézier-based Surrogate model with NAS-Bench-X11 and parametric learning curve model on CIFAR-10 and TinyImageNet. "GT (1 seed)" means a 1-seed tabular benchmark.

| Methods | CIFAR10 | | | | TinyImageNet | | | |
|---|---|---|---|---|---|---|---|---|
| | Avg.R2 | Final R2 | Avg.KT | Final KT | Avg.R2 | Final R2 | Avg.KT | Final KT |
| *Parametric learning curve neural network (Domhan et al., 2015)* | | | | | | | | |
| exp3 | 0.397 | 0.791 | 0.769 | 0.789 | -1.128 | 0.935 | 0.807 | 0.849 |
| ilog2 | 0.799 | 0.830 | 0.820 | 0.830 | 0.297 | 0.978 | 0.879 | 0.915 |
| pow2 | 0.212 | -0.056 | 0.564 | 0.506 | 0.321 | 0.396 | 0.571 | 0.547 |
| log power | 0.195 | 0.583 | 0.544 | 0.519 | -1.933 | 0.872 | 0.807 | 0.873 |
| logx linear | 0.808 | 0.825 | 0.810 | 0.793 | 0.779 | 0.969 | 0.893 | 0.906 |
| vapor | 0.790 | 0.671 | 0.830 | 0.829 | 0.897 | 0.957 | 0.858 | 0.883 |
| *NAS-Bench-X11 (Yan et al., 2021)* | | | | | | | | |
| SVD-XGB | 0.762 | 0.731 | 0.827 | 0.836 | 0.890 | 0.897 | 0.848 | 0.862 |
| SVD-LGB | 0.838 | 0.850 | 0.787 | 0.795 | 0.967 | 0.976 | 0.899 | 0.908 |
| SVD-MLP | 0.869 | 0.835 | 0.859 | 0.852 | 0.967 | 0.972 | 0.913 | 0.919 |
| BCS(Ours) | **0.892** | **0.872** | 0.860 | 0.841 | 0.968 | **0.979** | 0.922 | 0.928 |
| GT (1 seed) | 0.857 | 0.821 | **0.928** | **0.931** | **0.979** | 0.975 | **0.961** | **0.962** |

"GT" row in Table 3). The sampled configurations on CIFAR10 and TinyImageNet are split into training, validation, and testing sets containing 11597, 1288, and 1000 samples respectively.

**Evaluation Metrics.** Following Wen et al. (2020), White et al. (2021b) and Zela et al. (2022), we use the coefficient of determination $R^2$ and the Kendall rank correlation coefficient $\tau$ as the evaluation metrics. These two metrics only evaluate the performance based on the overall statistics of the curve rather than anomalies. However, a few spike anomalies on a validation curve could significantly affect the final accuracy prediction. As a result, we further adopt spike anomalies (Yan et al., 2021) as extra metrics ( detailed descriptions in appendix).

**Evaluation Results.** The performance of surrogate models is shown in Table 3. First, the parametric learning curve-based models function can not well fit the real learning curve in EA-HAS-Bench, and some of the higher order functions even fail to converge, such as $pow_3$ $(c - ax^{-\alpha})$ and $pow_4$ $(c - (ax + b)^{-\alpha})$. This is because the importance of the different parameters in a surrogate model varies considerably, especially the parameter which is in the exponent of an exponential function. The percentage of spike anomalies for real vs. BCS is 3.72% and 4.68% on CIFAR-10 and 0.83% and 1.31% on TinyImageNet, respectively. We further evaluate the consistency between the real measured energy cost and the predicted energy cost by the surrogate model. Specifically, on CIFAR-10, the energy cost surrogate model achieves R2 of 0.787, KT of 0.686, and Pearson correlation of 0.89. On TinyImageNet, it achieves R2 of 0.959, KT of 0.872, and Pearson correlation of 0.97.

## 3 DIFFERENCE WITH EXISTING NAS BENCHMARKS

Compared with existing NAS benchmarks such as NAS-Bench-101 (Ying et al., 2019), NAS-Bench-201 (Dong & Yang, 2020), NAS-Bench-301 (Siems et al., 2020) or Surrogate NAS Bench (Zela et al., 2022), EA-HAS-Bench has three significant differences.

**Diverse and Large Scale Joint Search Space.** EA-HAS-Bench is more diverse in terms of the types of configurations in the search space, which contains both model architectures and training hyperparameters. Although NAS-HPO-Bench (Klein & Hutter, 2019) and NAS-HPO-Bench-II (Hirose et al., 2021) also consider both architectures and hyperparameters, both benchmarks are based on small and homogeneous search spaces. Specifically, NAS-HPO-Bench focuses only on 2-layer feed-forward network training on tabular data, and the hyperparameters search space of NAS-HPO-Bench-II only contains learning rate and batch size. Besides the search space, diversity is also reflected in the evaluated performance ranges. As shown in Figure 2, the performance range of DARTS (Liu et al., 2019) used by NAS-Bench-301 is the smallest for the validation performance on CIFAR-10. Although DARTS contains more architectures, the performance of the models in this space is significantly less diverse (Yang et al., 2020). Compared with NAS-Bench-101/201, the configurations in EA-HAS-Bench cover a much larger performance range.

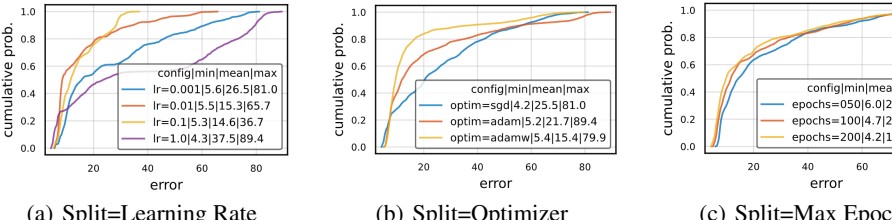

(a) Split=Learning Rate          (b) Split=Optimizer          (c) Split=Max Epoch

Figure 3: The empirical cumulative distribution (ECDF) of all real measured configurations on TinyImageNet for 3 different splits.

**Modeling Learning Curves for complex joint space.** To the best of our knowledge, NAS-Bench-X11 (Yan et al., 2021) is the only existing surrogate model that provides the full training status over the entire training process. However, NAS-Bench-X11 is only available for learning curves with fixed maximum epochs. The number of training epochs required for convergence is not always the same for different architectures and it also directly affects the training energy cost. As a result, for a more realistic search space like EA-HAS-Bench, we propose BSC to predict learning curves with different maximum epochs.

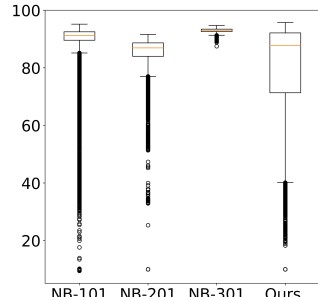

Figure 2: Validation accuracy box plots for each NAS benchmark in CIFAR-10. The whiskers represent the minimum and maximum accuracies in each search space. The black hollow circles represent outliers

**Full-cycle Energy Consumption.** Most existing benchmarks use model training time as the training resource budget, which is not an accurate estimation of energy cost. Firstly, the GPU does not function at a stable level throughout the entire training process, or it may not even be working at all for a significant time period during training, and hence longer training time does not always mean more GPU usage and more energy cost. Secondly, energy cost not only corresponds to training time but also relates to the energy cost per unit time, which is affected by the architecture and hyperparameter configurations. Our experimental analysis in the next section also verifies that training time and energy cost are not equivalent. HW-NAS-Bench (Li et al., 2021b) also considers energy consumption, but its focus is on the model inference energy cost. On the other hand, EA-HAS-Bench provides a full-cycle energy consumption, both during training and inference. The energy consumption metric allows HPO algorithms to optimize for accuracy under an energy cost limit (Section 5).

## 4 ANALYSIS ON EA-HAS-BENCH

**Impact of Hyperparameter Configurations.** Since most existing large-scale computer vision NAS benchmarks focus solely on network architectures and apply the same hyperparameters for all models, we examine how different hyperparameters affect the searched model performance. We use a *empirical cumulative distribution* (ECDF) (Radosavovic et al., 2020) to assess the quality of search space. Specifically, we take a set of configurations from the search space and characterize its error distribution. Figure 3 shows the empirical cumulative distribution (ECDF) of different training hyperparameters on CIFAR-10 and TinyImageNet. (The full version is in the Figure 13 of Appendix.) We observe that the learning rate and the choice of optimizer may have the most significant impact on the search space quality. The size of the maximum number of epochs is also positively correlated to the quality of the search space.

**Correlation Between Training Energy Cost and Training time.** Figure 4 investigates the relationship between the training energy consumption (TEC), training time, and the test accuracy of models in TinyImageNet. Firstly, we observe that the points in the left figure of Figure 4 is rather scattered. This means *the correlation between training time and energy cost is not strong.* Although training the model for a longer period is likely to yield a higher energy cost, the final cost still depends on many other factors including power (i.e., consumed energy per hour). The middle and right plots of Figure 4 also verifies the conclusion, where the models in the Pareto Frontier on the

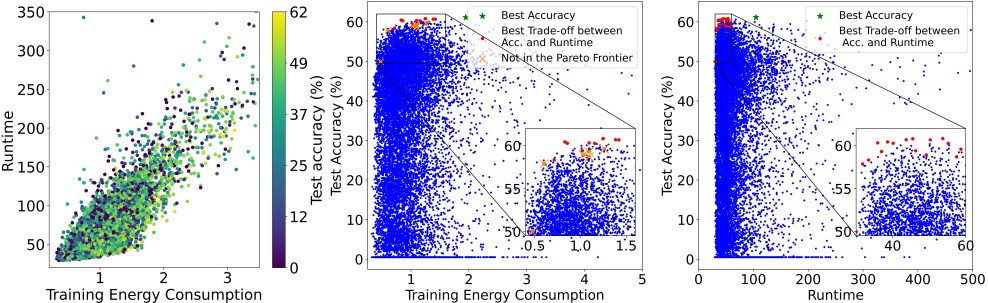

Figure 4: **(left)** Training time vs. training energy consumption (TEC), color-coded by test accuracy. **(middle)** Test Accuracy vs. TEC. **(right)** Test Accuracy vs. training time. TEC and training time are the per epoch training energy consumption (Kw*h) and runtime (seconds) on the Tesla V100. The orange cross in the middle plot denotes the models in the Pareto Frontier on the accuracy-runtime coordinate but are not in the Pareto Frontier on the accuracy-TEC.

accuracy-runtime coordinate (right figure) are not always in the Pareto Frontier on the accuracy-TEC coordinate (middle figure), showing that training time and energy cost are not equivalent.

Meanwhile, training a model longer (or with more energy) does not guarantee better accuracy. In the middle and right plots of Figure 4, we see that many models with high TECs, still fail to train, due to improper neural architectures or hyperparameter configurations. On the other hand, simply searching for the best accuracy might not be cost-efficient, since there are quite a few configurations with the same level of accuracy. The observation motivates finding the best trade-off between accuracy and energy consumption.

**Configurations–Accuracy/Energy Correlation.** Figure 5 shows the correlation between architecture/ hyperparameter configurations and accuracy, runtime, TEC, and inference energy cost (IEC). We observe that hyperparameters like learning rate also have a high correlation with model performance, which further verifies the importance to consider hyperparameters in NASBench. Both network architecture like network depth and width and hyperparameters like training epoch has a relatively strong correlation with energy cost, showing the importance of considering both perspectives in our energy-aware Benchmark.

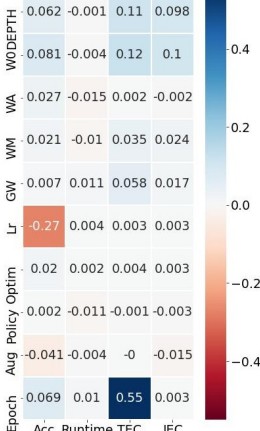

Figure 5: Correlation coefficient between RegNet+ HPO and Accuracy, Runtime, TEC, and inference energy cost (IEC) on TinyImageNet.

## 5   EA-HAS-BENCH AS A BENCHMARK

EA-HAS-Bench saves tremendous resources to train and evaluate the configurations for real. We demonstrate how to leverage the proposed dataset to conduct energy-aware AutoML research with two use cases. Firstly, we evaluate the trade-off between search energy cost and model performance of four single-fidelity algorithms: random search (RS) (Li & Talwalkar, 2019), local search (LS) (White et al., 2020), regularized evolution (REA) (Real et al., 2019), BANANAS (White et al., 2021a), and two multi-fidelity bandit-based algorithms: Hyperband (HB) (Li et al., 2017) and Bayesian optimization Hyperband (BOHB) Falkner et al. (2018). The implementation details of the above algorithms are in Appendix D. Then, in the second sub-section, as the first step toward a long journey of energy-aware AutoML, we arm several existing AutoML algorithms with another energy-related objective and verify the effectiveness of energy-aware AutoML on our hyperparameter-architecture benchmark.

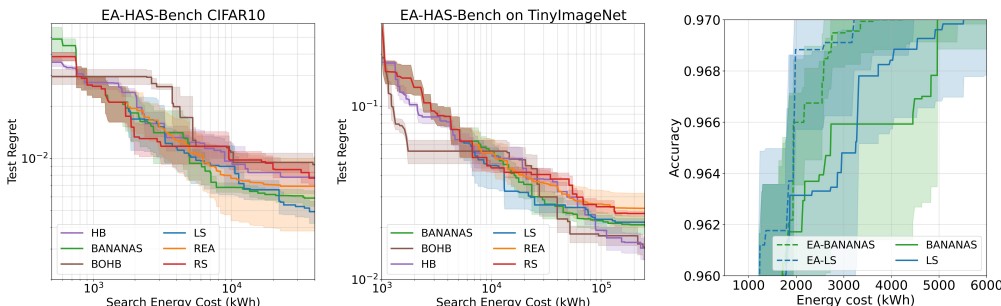

Figure 6: NAS results on CIFAR10 (**left**) and TinyImageNet (**middle**). Energy-aware BANANAS and Energy-aware Local Search (LS) vs. origin BANANAS and LS on CIFAR10 (**right**).

### 5.1 BENCHMARKING EXISTING ALGORITHMS

**Experimental setup.** Since EA-HAS-Bench focuses on the trade-off between model performance and search energy cost, in this experiment we use the total search energy cost as the resource limitation, instead of training time. As a result, we set the maximum search cost to roughly 40,000 kWh for CIFAR 10 and 250,000kWh for TinyImageNet, which is equivalent to running a single-fidelity HPO algorithm for about 1,000 iterations.

**Results.** In Figure 6 left and middle, we compare single- and multi-fidelity algorithms on the search space of EA-HAS-Bench. For single-fidelity algorithms, LS is the top-performing algorithm across two datasets. This shows that similar to NAS-Bench, HAS-Bench with a joint search space also has locality, a property by which "close" points in search space tend to have similar performance.

### 5.2 A NEW ENERGY-AWARE HPO BASELINE

**Energy Aware HPO** Most existing HPO methods do not directly consider the search energy cost. In order to verify the importance to consider the energy cost during the search process, we propose a new energy-aware HPO baseline by modifying existing HPO methods. Following MnasNet (Tan et al., 2019), we modify the BANANAS (White et al., 2021a) and LS (White et al., 2020) by changing the optimization goal to a metric that considers both accuracy and energy cost: $ACC \times \left(\frac{TEC}{T_0}\right)^w$, where $T_0$ is the target TEC. For CIFAR10, we set the $T_0$ to 0.45 and $w$ to -0.07.

**Experimental Setup and Result** We explore another important usage scenario, where the goal is to achieve a target model performance using as little energy cost as possible. As a result, we use model performance rather than energy consumption as a resource limitation and stop the search when the model hits a target performance and compare the corresponding search energy cost. For CIFAR10, the target accuracy is set to 97%. As shown in Figure 6 right, EA algorithms that consider TEC save close to 20% of search energy consumption compared to the origin algorithms in achieving the target accuracy. The ablation study is shown in Appendix D.4.

## 6 CONCLUSION

EA-HAS-Bench is the first large-scale energy-aware hyperparameter and architecture search benchmark. The search space of EA-HAS-Bench consists of both network architecture scale and training hyperparameters, covering diverse configurations related to energy cost. A novel Bézier curve-based surrogate model is proposed for the new joint search space. Furthermore, we analyze the difference between existing NAS benchmarks and EA-HAS-Bench and dataset statistics and the correlation of the collected data. Finally, we provide use cases of EA-HAS-Bench to show that energy-aware algorithms can save significant energy in the search process. We expect that our EA-HAS-Bench expedites and facilitates the EA-NAS and HAS research innovations.

## 7 ACKNOWLEDGMENTS AND DISCLOSURE OF FUNDING

S. Dou and C. Zhao acknowledge that the work was supported by the National Natural Science Fund of China (62076184, 61976158, 61976160, 62076182, 62276190), in part by Fundamental Research Funds for the Central Universities and State Key Laboratory of Integrated Services Networks (Xidian University), in part by Shanghai Innovation Action Project of Science and Technology (20511100700) and Shanghai Natural Science Foundation (22ZR1466700). We thank Yuge Zhang (Microsoft Research Asia) for suggestions on the design of the search space and for revising the writing of the paper. We thank Bo Li (Nanyang Technological University) for help with the codes.

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

## A    RELATED WORK

### A.1    EXISTING NAS BENCHMARKS

**Tabular NAS Benchmarks.**  While neural architecture search (NAS) has succeeded in various practical tasks such as image recognition (Pham et al., 2018) and sequence modeling (So et al., 2019), the non-reproducibility of NAS has been contested (Li & Talwalkar, 2019). One of the main reasons that complicate NAS reproducibility studies is the high computational cost as well as the large carbon emissions that result(Li et al., 2021a). For the reproducibility of NAS research, NAS-Bench-101 (Ying et al., 2019) and NAS-Bench-201 (Dong & Yang, 2020) have been proposed. Due to the success of these tabular benchmarks in image classification, corresponding benchmarks have been proposed in areas such as NLP and speech recognition (Mehta et al., 2022).

**Surrogate NAS Benchmarks.**  The recent Surrogate NAS Benchmarks (Zela et al., 2022) builds surrogate benchmarks on a realistic search space and shows strong generalization performance. To extend the surrogate model, NAS-Bench-X11 (Yan et al., 2021) uses singular value decomposition and noise modeling to output the learning curve.

**NAS Benchmarks joint HPO**  Most NAS Benchmarks have fixed training hyperparameters, but the training hyperparameters strongly influence the performance of the model obtained by NAS (Dong et al., 2020). To alleviate this problem, NAS-HPO-Bench (Klein & Hutter, 2019) and NAS-HPO-Bench-II Hirose et al. (2021) are proposed. However, as shown in Table 1, the size of the two Benchmarks is small and the search spaces are simple. The architecture space of NAS-HPO-Bench is multi-layer perception (MLP) trained on the tabular datasets for regression tasks. NAS-HPO-Bench-II only really measures 12 epochs using the CIFAR-10 dataset Krizhevsky et al. (2009) and the training hyperparameters space only contains learning rate and batch size. A similar work to ours is JAHS-Bench-201 (Bansal et al.), also with a large-scale joint search space. JAHS-Bench-201 provides FLOPS, latency, and runtime in addition to performance and loss. However, JAHS-Bench-201 does not focus on energy consumption during the search.

### A.2    RESOURCE-AWARE NAS

Early NAS algorithms (Zoph & Le, 2016; Real et al., 2019) focused only on performance and ignored some of the associated hardware consumption. To this end, many resource-aware NAS algorithms are proposed to balance performance and resource budgets (He et al., 2021). These resource-aware NAS algorithms focus on the four types of computational costs that are included the FLOPs, parameter size, the number of Multiply-ACcumulate (MAC) operations, and real latency. Among the NAS algorithms, two classic works are MnasNet (Tan et al., 2019) and FBNet (Wu et al., 2019). MnasNet proposes a multi-objective neural architecture search approach that optimizes both accuracy and real-world latency on mobile devices. Similar to MansNet, FBNet designs a loss function to trade off the cross-entropy loss and latency of architecture. However, none of the above NAS algorithms focus on the huge energy consumption in search.

## B    MORE DETAILS OF SECTION 2

### B.1    MORE DETAILS ON EVALUATION METRICS

**Details of Spike Anomalies.** Although R2 and KT can evaluate the surrogate model by measuring the overall statistics between the surrogate benchmark and the ground truth, they are not sensitive to anomalies. Following, NAS-bench-X11 Yan et al. (2021), to evaluate the performance of surrogate models based on anomalies, we introduce the Spike Anomalies metrics. We first calculate the largest value $x$ such that there are fewer than 5% of learning curves whose maximum validation accuracy is higher than their final validation accuracy, on the true learning curves. Next, the percentage of surrogate learning curves whose maximum validation accuracy is $x$ higher than their final validation accuracy was computed.

B.2    MORE DETAILS ON BÉZIER CURVE-BASED SURROGATE MODEL

**Network Structure of BSC.**    The encoder of architecture and hyperparameters adopts a simple Multi-Layer Perceptron (MLP) structure, consisting of two linear layers with ReLU activation functions. Then, the encoded features are fed into the learning curve prediction network, which is also an MLP with an extra dropout layer, whose output is fed into two linear regressors that output the coordinates of the control points. We use the sigmoid activation function to the regressor, which directly constrains the initial final performance between 0 and 1. The control points are learned with the standard least square method as follows:

$$
\begin{bmatrix}
B_{0,5}(t_0) & \cdots & B_{5,5}(t_0) \\
B_{0,5}(t_1) & \cdots & B_{5,5}(t_1) \\
\vdots & \ddots & \vdots \\
B_{0,5}(t_m) & \cdots & B_{5,5}(t_m)
\end{bmatrix}
\begin{bmatrix}
i_{x_{start}} & i_{y_{start}} \\
c_{x_1} & c_{y_1} \\
\vdots & \vdots \\
i_{x_{end}} & i_{y_{end}}
\end{bmatrix}
=
\begin{bmatrix}
e_{x_0} & e_{y_0} \\
e_{x_1} & e_{y_1} \\
\vdots & \vdots \\
e_{x_m} & e_{y_m}
\end{bmatrix}
\tag{4}
$$

**Details on the Compared Parametric Learning Curves based Methods.**    Several parametric learning curves-based methods are selected as the comparison methods. The detailed formulation of those parametric models is shown in Table 4. Following Domhan et al. (2015), we first try the Levenberg-Marquardt algorithm and fall back to Broyden–Fletcher–Goldfarb–Shanno (BFGS) in case that fails. In the experiment, we found that the initial parameters are important in the fitting process. Some parametric models of a high order cannot be fitted to the learning curve in the training set because suitable initial parameters cannot be found. In contrast, the initial point of the BSC model is the starting and ending point of the learning curve.

Table 4: The formula of parametric learning curve

| Reference name | Formula |
| --- | --- |
| $\exp_3$ | $c - exp(-ax + b)$ |
| $ilog_2$ | $c - \frac{a}{\log x}$ |
| $pow_2$ | $ax^{\alpha}$ |
| log power | $\frac{a}{1+\left(\frac{x}{e^b}\right)^c}$ |
| logx linear | $a log(x) + b$ |
| vapor | $\exp\left(a + \frac{b}{x} + c \log(x)\right)$ |

**Hyperparameters of Surrogate Models.**    Table 3 shows the optimal hyper-parameters searched by TPE for different surrogate models. Due to the page limit, here we only listed the hyperparameters of the three models that achieve the best performance in Table 5. We used a fixed budget of 500 trials for all surrogate models and average R2 as the optimal target.

**Surrogate Models of Runtime Metrics**    Although our benchmark focuses on energy consumption, we also provide runtimes to allow NAS methods to use runtimes as budgets. We train an LGB model with the average runtime of each epoch as a target and the model achieves 0.926 for R2 and 0.849 for KT on runtime prediction on CIFAR10.

B.3    MORE DETAILS ON DATA COLLECTION

Here we provide a more detailed introduction to energy consumption measurement for data collection. Intuitively, the search energy cost is the total energy consumption to complete a search algorithm. Since the majority of the energy cost comes from training each deep model the search algorithm traverses, in our dataset, the search energy cost is defined as the total energy cost (in kWh) or time (in seconds) to train the model configurations traversed by the search algorithms.

Specifically, we denote a training configuration in the EA-HAS-Bench search space as $\mathbf{c} \in N^d$, where $\mathbf{c}$ is a $d$-dimensional vector containing $d$ training parameters. $e_{ep}(\mathbf{c})$ is the energy cost measure function that returns the training energy cost to train a model with training configuration $\mathbf{c}$ for one epoch. $A = \{\mathbf{c}^{(i)}\}_{i=0}^{N}$ is the set of configurations a NAS/HPO search method traversed. As a

Table 5: Hyperparameters of the surrogate models and the optimal values found via TPE.

| Model | Hyperparameter | Range | Type | Optime Value |
|---|---|---|---|---|
| SVD-LGB | Num. components | [1,20] | uniform int | 4 |
| | Num. rounds | - | constant | 3000 |
| | Early Stopping | - | constant | 100 |
| | Max. depth | [1,24] | uniform int | 9 |
| | Num. leaves | [10, 100] | uniform int | 84 |
| | Min. child weight | [0.001, 10] | log uniform | 0.4622 |
| | Lambda L1 | [0.001, 1000] | log uniform | 0.0056 |
| | Lambda L2 | [0.001, 1000] | log uniform | 0.0054 |
| | Boosting type | - | constant | gbdt |
| | Learning rate | [0.001, 0.1] | log uniform | 0.5822 |
| SVD-MLP | Num. components | [1,20] | uniform int | 3 |
| | Num. epochs | [5,200] | uniform int | 190 |
| | hidden dim | [32, 250] | constant | 183 |
| | Num. layers | - | constant | 4 |
| | Learning rate | [0.0001, 0.1] | log uniform | 0.0008 |
| | drop out | [0.0, 0.1, 0.2, 0.3, 0.4, 0.5] | uniform int | 0.2 |
| | batch size | [64, 128, 256] | uniform int | 64 |
| BSC | Num. epochs | [5,200] | uniform int | 180 |
| | hidden dim | [32, 250] | constant | 247 |
| | Num. layers | - | constant | 4 |
| | Learning rate | [0.0001, 0.1] | log uniform | 0.0003 |
| | drop out | [0.0, 0.1, 0.2, 0.3, 0.4, 0.5] | uniform int | 0 |
| | batch size | [64, 128, 256] | uniform int | 64 |
| LGB-E | Num. rounds | - | constant | 3000 |
| | Early Stopping | - | constant | 100 |
| | Max. depth | [1,100] | uniform int | 31 |
| | Num. leaves | [10, 1000] | uniform int | 315 |
| | Min. child weight | [0.001, 10] | log uniform | 0.0046 |
| | Lambda L1 | [0.001, 1000] | log uniform | 20.3216 |
| | Lambda L2 | [0.001, 1000] | log uniform | 11.6866 |
| | Boosting type | - | constant | gbdt |
| | Learning rate | [0.001, 0.1] | log uniform | 0.0451 |

result, the total search energy $e_s$ cost is defined as:

$$e_s(A) = \sum_{\mathbf{c}^{(i)} \in A} e_{ep}(\mathbf{c}^{(i)}) * \mathbf{c}_n^{(i)}, \tag{5}$$

where $n$ is the index of $\mathbf{c}$ that stores the number of total training epochs to train the deep model under configuration $\mathbf{c}$.

Next, we introduce how to measure the per epoch energy consumption for different training configurations. Following Li et al. (2021a), we collect the real-time power of the GPU during the operation of the algorithm through the interface of pynvml. In the following, we will provide an implementation of the GPU tracker to accurately describe its functionality.

```python
import re
import subprocess
import threading
import time
import pynvml

import torch
import xmltodict

class Tracer(threading.Thread):
    def __init__(self, gpu_num=(0,), profiling_interval=0.1):
        threading.Thread.__init__(self, )
        ...

    def run(self):
        pynvml.nvmlInit()
        handle = pynvml.nvmlDeviceGetHandleByIndex(0)
        power_list = []
        while self._running:
```

```
20            self.counters += 1
21            power_u_info = pynvml.nvmlDeviceGetPowerUsage(handle)
22            power_list.append(power_u_info/1000)
23            time.sleep(self.profiling_interval)
24
25 class GPUTracer:
26     all_mode =['normal']
27     def __init__(self, mode, gpu_num=(0,), profiling_interval=0.1,
       verbose=False):
28         if not mode in GPUTracer.all_modes:
29             raise ValueError(f'Invalid mode : {mode}')
30         self.mode = mode
31         self.gpu_num = gpu_num
32         self.profiling_interval = profiling_interval
33         self.verbose = verbose
34
35     def wrapper(self, *args, **kwargs):
36         if not GPUTracer.is_enable:
37             return self.func(*args, **kwargs), None
38         tracer = Tracer(gpu_num=self.gpu_num, profiling_interval=self.
       profiling_interval)
39         start = torch.cuda.Event(enable_timing=True)
40         end = torch.cuda.Event(enable_timing=True)
41         start.record()
42         tracer.start()
43         results = self.func(*args, **kwargs)
44         tracer.terminate()
45         end.record()
46         torch.cuda.synchronize()
47
48         if tracer.counters == 0:
49             print("*" * 50)
50             print("No tracing info collected, increasing sampling rate if
        needed.")
51             print("*" * 50)
52             tracer.join()
53             return results, None
54         else:
55             tracer.join()
56             avg_power, avg_temperature, avg_gpu_utils, avg_mem_utils,
       total_power, total_gpu_utils, total_mem_utils = tracer.communicate()
57             time_elapse = start.elapsed_time(end) / 1000
58             energy_consumption = time_elapse * avg_power / 3600
```

Listing 1: GPU Tracer

Specifically, we implemented this tracer using Python's decorator function and then just logged the GPU information at runtime. In the following, we provide a user case for collecting energy data.

```
1 @GPUTracer(mode='normal', verbose=True)
2 def train_epoch(loader, model, ...):
3     """Performs one epoch of training."""
4     ...
5
6 @GPUTracer(mode='normal', verbose=True)
7 @torch.no_grad()
8 def test_epoch(loader, model, ....):
9     """Evaluates the model on the test set."""
```

Listing 2: GPU information Collection by GPUTracer

The details of the machines used to collect energy consumption are in Table 6.

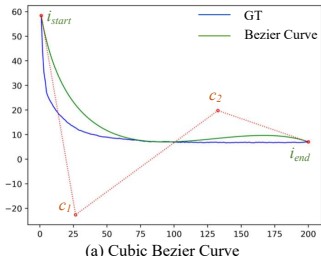 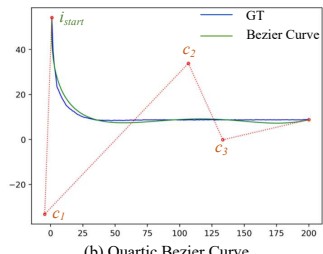 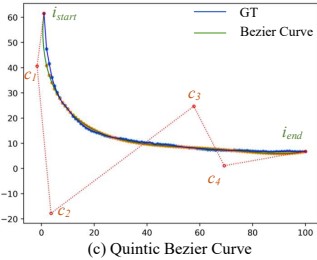

(a) Cubic Bezier Curve          (b) Quartic Bezier Curve          (c) Quintic Bezier Curve

Figure 7: Visualization of generated Bézier Curves and original learning curves (Ground Truth).

Table 6: Details of the machines used to collect energy consumption

| Property Name | Value |
|---|---|
| CPU | Intel(R) Xeon(R) CPU E5-2690 v3 @ 2.60GHz 2600 MHz |
| Memory-GB | 112 |
| Operation system | Linux Ubuntu 20.04 LTS |
| Hard drive-GB | 1000 |
| GPU | Nvidia Tesla V100 with 32 GB memory |

### B.4 MORE ABLATION STUDY ON SURROGATE MODEL

**The effect of Degree** $n$    In Table 7, we show the effect of different degrees of the Bézier curve on the prediction performance. The higher the degree, the better the Bézier curve fits the real learning curve, but it also leads to overfitting. As the degree increases, the prediction performance becomes worse instead. When the Degree is 4, the Quadratic Bézier curve achieves the best results.

**Bézier curves vs. Polynomial functions**    Compared to general $n$ order polynomial functions, the coefficients of the Bézier Curve are explainable and have real-world semantics (i.e. the control points that define the curvature). As a result, we can leverage the prior knowledge of the learning curve by adding constraints to the control points and fitting a better learning curve. For example, in our implementation, we constrained the starting and ending points of the learning curve to make the accuracy value stay within the $[0, 1]$ range.

Empirically, we conduct an ablation study in which instead of predicting the Bézier Curve control points, directly predicts the coefficients and intercept of polynomial functions. However, we observe that for polynomial functions of higher order (n=4), the model is almost impossible to fit. The possible reason is that the scales of the parameters differ too much, and the magnitude of the coefficients varies widely, making it difficult to learn the model. When we set n to 2, the results are as shown in Table 8. In contrast, regardless of the order of Bézier's curve, the size of the control points is basically in the same order of magnitude and the model can be easily fitted (as shown in Table 7)

### B.5 API ON HOW TO USE EA-HAS-BENCH

Here is an example of how to use EA-HAS-Bench:

```
def get_ea_has_bench_api(dataset):
    full_api = {}
    # load the ea-nas-bench surrogate models
    if dataset=="cifar10":
        ea_has_bench_model = load_ensemble('checkpoints/ea_has_bench-v0.2')
        train_energy_model = load_ensemble('checkpoints/ea_has_bench-trainE-v0.2')
        test_energy_model = load_ensemble('checkpoints/ea_has_bench-testE-v0.1')
```

Table 7: The predicted performance of different degrees of the Bézier Curve on CIFAR-10

| Method | Avg R2 | Final R2 | Avg KT | Final KT |
|---|---|---|---|---|
| Cubic Bézier (Degree = 3) | 0.870 | 0.859 | 0.858 | **0.855** |
| Quadratic Bézier (Degree = 4) | **0.892** | **0.872** | **0.860** | 0.841 |
| Quintic Bézier (Degree = 5) | 0.891 | 0.862 | 0.855 | 0.834 |
| Sextic Bézier (Degree = 6) | 0.852 | 0.796 | 0.771 | 0.736 |

Table 8: The polynomial function model on CIFAR10

| Degree | Avg.R2 | Avg.KT | Final.R2 | Final.KT |
|---|---|---|---|---|
| n=2 | 0.0437 | 0.547 | -2.66 | 0.182 |

```
9       runtime_model = load_ensemble('checkpoints/ea_has_bench-runtime-
    v0.1')
10    elif dataset=="tiny":
11        ea_has_bench_model = load_ensemble('checkpoints/ea-nas-bench-tiny
    -v0.2')
12        train_energy_model = load_ensemble('checkpoints/ea-nas-bench-
    trainE-v0.1')
13        test_energy_model = load_ensemble('checkpoints/ea-nas-bench-testE
    -v0.1')
14        runtime_model = load_ensemble('checkpoints/ea-nas-bench-runtime-
    v0.1')
15
16    full_api['ea_has_bench_model'] = [ea_has_bench_model, runtime_model,
    train_energy_model, test_energy_model]
17    return full_api
18
19 ea_api = get_ea_has_bench_api("cifar10")
20
21 # output the learning curve, train time, TEC and IEC
22 lc = ea_api['ea_has_bench_model'][0].predict(config=arch_str)
23 train_time = ea_api['ea_has_bench_model'][1].predict(config=arch_str)
24 train_cost = ea_api['ea_has_bench_model'][2].predict(config=arch_str)
25 test_cost = ea_api['ea_has_bench_model'][3].predict(config=arch_str)
```

Listing 3: EA-HAS-Bench API

## C  MORE ANALYSIS ON EA-HAS-BENCH

### C.1  MORE ANALYSIS ON IMPACT OF HYPERPARAMETER CONFIGURATIONS

Table 13 compares how different hyper-parameter configurations affect the search space quality on both CIFAR10 and TinyImageNet datasets. Besides the learning rate, optimizer, and the number of total training epochs discussed in the main paper, here we further examine the influence of learning rate policy and data augmentation. For the investigation of data augmentation, we found that cutout achieves opposite effects on different datasets.

### C.2  COMPARING WITH ORIGINAL REGNET SPACE

As shown in the figure 8, we compare the learning curves of one hundred samples actually measured from under two search spaces-RegNet and RegNet+HPO. The different architectures converge effectively under well-designed training hyperparameters. However, many architectures fail to converge under the search space of RegNet+HPO. But the latter may find a better combination to produce better performance. With a fixed training hyperparameter, the best model obtained by searching only in the network architecture space is not necessarily the true best model. At these fixed training hyperparameters, the sub-optimal model obtained by searching may get better performance under another set of training hyperparameters. In addition, the optimal hyperparameters are difficult

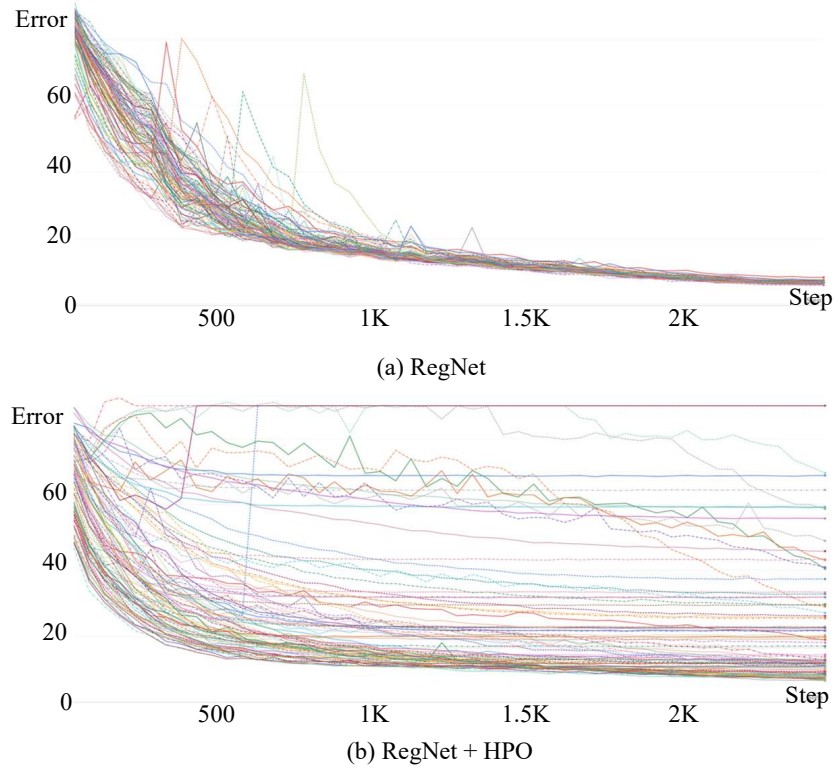

Figure 8: Comparing the learning curve under RegNet and RegNet+HPO search spaces. (a) The RegNet search space. (b) The RegNet + HPO search space.

to determine when facing a new dataset, while searching for training hyperparameters and model architecture is a more realistic scenario.

### C.3 MORE ANALYSIS ON ENERGY COST

**Power Distribution** The correlation between energy consumption and runtime is $energy = runtime \times Avg.power$. As shown in Figure 9, since the power of different models trained on Tesla V100 is not constant, the energy consumption is not linearly related to the runtime.

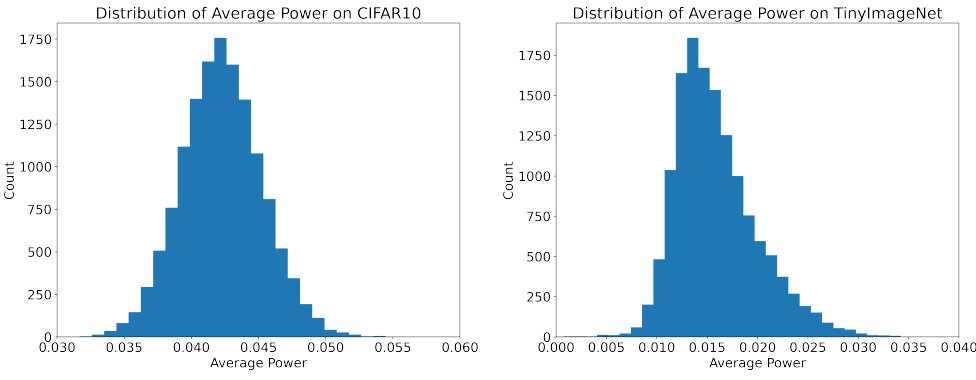

Figure 9: Visualization of distributions of power on CIFAR10 (left) and TinyImageNet (right).

**Correlation Between Training Energy Cost and FLOPs.** Although HW-NAS-Bench has discussed the relationship between FLOPs and energy in detail on six different hardware devices and concluded that the theoretical hardware consumption does not always correlate with hardware consumption in both search spaces, we still analyze the correlation between the two in our search space. The Kendall Rank Correlation Coefficient between FLOPs and TEC on TinyImageNet is 0.396 which is less than 0.5. As shown in Figure 10, we observe that the distribution of scattered points is triangular rather than linear. Based on the qualitative and quantitative results, the correlation between FLOPs and TEC is not strong.

**Overall Energy Consumption of Sampled Configurations** The total energy consumption to build EA-HAS-Bench in Table 9.

Table 9: The energy consumption (kWh) to build EA-HAS-Bench

| Dataset | Tranining & Validation & Testing sets | GT(1 seed) | Total |
|---------|---------------------------------------|------------|-------|
| CIFAR10 | 660,313k | 46,813 | 707,126 |
| TinyImageNet | 1,715,985 | 124,088 | 1,840,074 |
| Total | | | 2,547,200 |

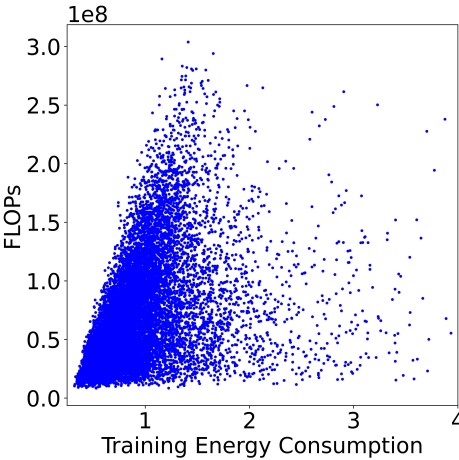

Figure 10: FLOPs vs. training energy consumption(TEC) on TinyImageNet.

# D   MORE DETAILS OF SECTION 5

## D.1   INTRODUCTION ON COMPARED NAS/HPO METHODS

All algorithms we use in Section 5 are based on NASLib (Ruchte et al., 2020). For regularized evolution and local search, we modify them to suit our joint search space. The algorithm description and implementation details are as follows.

- **Random Search** is a classic baseline of HPO algorithms and is the basis for some complex algorithms (Li & Talwalkar, 2019). For EA-HAS-Bench, we sample randomly both the architecture and the hyperparameter space. The sampling type is the same as the dataset that we build for training the surrogate model.

- **Local search** iteratively evaluates all architectures in the neighborhood of the current best architecture found so far (White et al., 2020). For the search space of EA-HAS-Bench, we define *neighborhood* as having the same network architecture or the same training hyperparameters, i.e., points with the same subspace are neighboring points.

- **Regularized evolution** mutates the best architecture from a sample of all architectures evaluated so far. For EA-HAS-Bench, we define a *mutation* as a random change of one dimension in the architecture or hyperparameters.

- **BANANAS** is based on Bayesian optimization and samples the next point by acquisition function (White et al., 2021a). We used a modified version from NAS-Bench-X11 to construct the candidate pool by mutating the best four points 10 times each.

- **Hyperband** is based on the random search with successive halving (Li et al., 2017). Since the space size of EA-NAS-Bench is $3 \times 10^{10}$, we expect to explore more points and set the maximum budget to 512 and the minimum budget to 4.

- **Bayesian optimization Hyperband** is based on Hyperband with Bayesian optimization (Falkner et al., 2018). We use the same parameters as Hyperhand.

## D.2 MORE DETAILS ON EXPERIMENTAL SETUP

Following Ying et al. (2019) and Yan et al. (2021), during the architecture search we keep track of the best architectures found by the algorithm after each evaluation and rank them according to their validation accuracy. When the metric we specify (e.g., total energy consumption or target accuracy) exceeds the limit we set, we stop the search. After the search, we query the corresponding best accuracy of the model. We then compute regret:

$$regret_i = Acc_i - Acc^*$$ (6)

where $Acc_i$ denotes the accuracy of the best architecture after each evaluation $i$ and $Acc^*$ denotes the model with the highest average accuracy in the entire dataset. For experiments in section 5, we run 5 trials of each AutoML algorithm and compute the mean and standard deviation.

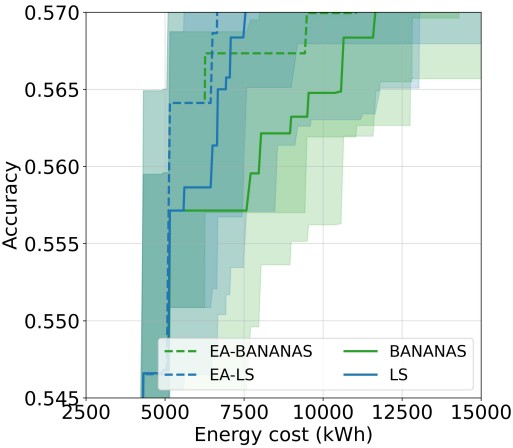

Figure 11: Energy-aware BANANAS and Energy-aware LS vs. origin BANANAS and LS on Tiny-Imaget.

## D.3 MORE RESULTS ON ENERGY-AWARE BASELINES

We also conduct experiments to evaluate the proposed energy-aware baselines on the TinyImageNet dataset. Specifically, we set T to 2.5 and target performance to 57%. The result is shown in Figure 11. Same to the experiment on CIFAR-10, compared with the baseline without the energy cost penalty, our EA algorithm costs significantly lower search energy to reach the target accuracy.

## D.4 ABLATION STUDY ON ENERGY-AWARE BASELINES

In this section, we conduct a more comprehensive empirical evaluation of the proposed new energy-aware baselines. Specifically, an ablation study is conducted to examine how different reward functions and hyper-parameters in the proposed objective affect the performance of the energy-aware HPO.

**Choice of the reward function.** Inspired by existing work on joint optimization of accuracy and latency (Tan et al., 2019; Bender et al., 2020), we modify the existing NAS methods (LS and BA-NANAS) by using multi-objective optimization functions, including soft exponential reward function (SoftE), hard exponential reward function (HardE), absolute reward function (Absolute). The soft exponential reward function is expressed as

$$r(\alpha) = Acc(\alpha) \times (T(\alpha)/T_0)^\beta \tag{7}$$

where $Acc(\alpha)$ denotes the accuracy of a candidate architecture $\alpha$, $T(\alpha)$ is its training energy consumption, $T_0$ denotes the TEC target to control the total TEC in the search process, and $\beta < 0$ is the cost exponent. The hard exponential reward function imposes a "hard" limit constraint:

$$r(\alpha) = \begin{cases} Acc(\alpha), & \text{if } T(\alpha) \leq T_0 \\ Q(\alpha) \times (T(\alpha)/T_0)^\beta, & \text{if } T(\alpha) > T_0 \end{cases} \tag{8}$$

The absolute reward function aims to find a configuration whose TEC is close to $T_0$:

$$r(\alpha) = Acc(\alpha) + \beta \left| T(\alpha)/T_0 - 1 \right| \tag{9}$$

For the three functions, we set TEC target $T_0 = 0.45$ and $\beta = -0.07$. In Table 10, we compare the total energy required by these methods to achieve the target performance. Since SoftE requires the least amount of energy, we choose SoftE to convert the current NAS algorithm to the EA-NAS algorithm.

Table 10: The total TEC in reaching target performance on CIFAR10

| Algorithms | Origin | Soft Exp | Hard Exp | Absolute |
|---|---|---|---|---|
| Local Search | 5,521 | 3,218 | 3,595 | 6,070 |
| BANANAS | 4,966 | 3,630 | 5,227 | 5,227 |

**Choice of the scaling.** Following MnasNet, we simply set the $beta$ to -0.07. Based on the TEC distribution of sampling points mainly between 0.4 and 0.5, we try three different parameters. The results are shown in the table. After $T_0 \geq 0.45$, the final Total TEC is the same, which indicates that EA-NAS is robust to the parameter $T_0$.

Table 11: The impact of different $T_0$ in CIFAR10

| Algorithms | $T_0 = 0.4$ | $T_0 = 0.45$ | $T_0 = 0.5$ |
|---|---|---|---|
| EA-BNANAS | 3,692 | 3,630 | 3,630 |
| EA-LS | 4,751 | 3,218 | 3,218 |

# E  SMALL REAL TABULAR BENCHMARK

Besides providing a large-scale proxy benchmark and the tens of thousands of sampling points used to construct it, we also provide a small real tabular benchmark. As shown in Table 12, we redefine a very small joint search space with a size of 500. As with the previous tabular benchmark, we evaluate all models within this space.

**NAS algorithms on Tabular Benchmark.** Similar to Section 5.1, we implement 6 NAS algorithms on the small tabular benchmark. In this experiment, the maximum search cost is set to 20,000 kWh, which is equivalent to running a single-fidelity HPO algorithm for about 200 iterations. We run 10 trials of each AutoML algorithm and compute the mean and standard deviation. The result is shown in Figure 12. Due to the different search spaces and budgets, the conclusions drawn differ slightly from the previous ones on the surrogate benchmark.

Table 12: Overview of the toy search space

| Type | Hyperparameter | Range | Quantize | Space |
|---|---|---|---|---|
| | Depth $d$ | [6,15] | 1 | 10 |
| | $w_0$ | [80, 112] | 8 | 5 |
| RegNet | $w_a$ | 20 | - | 1 |
| | $w_m$ | 2.75 | - | 1 |
| | Group Width | 16 | - | 1 |
| | Total of Network Architectures | | | 50 |
| | Learning rate | {0.001, 0.003, 0.005, 0.01, 0.03, 0.05, 0.1, 0.3, 0.5, 1.0} | - | 10 |
| Optim | Max epoch | {100} | - | 1 |
| | Decay policy | {'cos'} | - | 1 |
| | Optimizer | sgd | - | 1 |
| Training | Data augmentation | None | - | 1 |
| | Total of Hyperparameter Space | | | 10 |

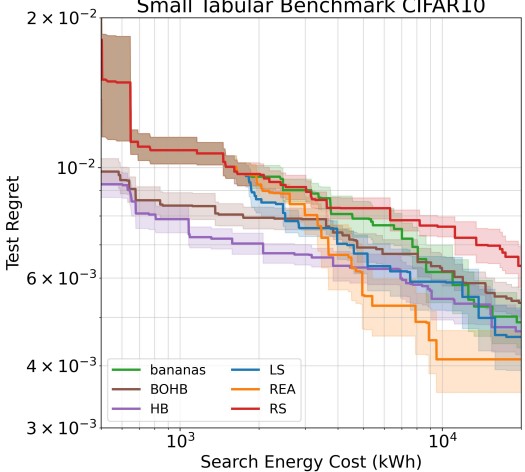

Figure 12: NAS results on small tabular benchmark CIFAR10 .

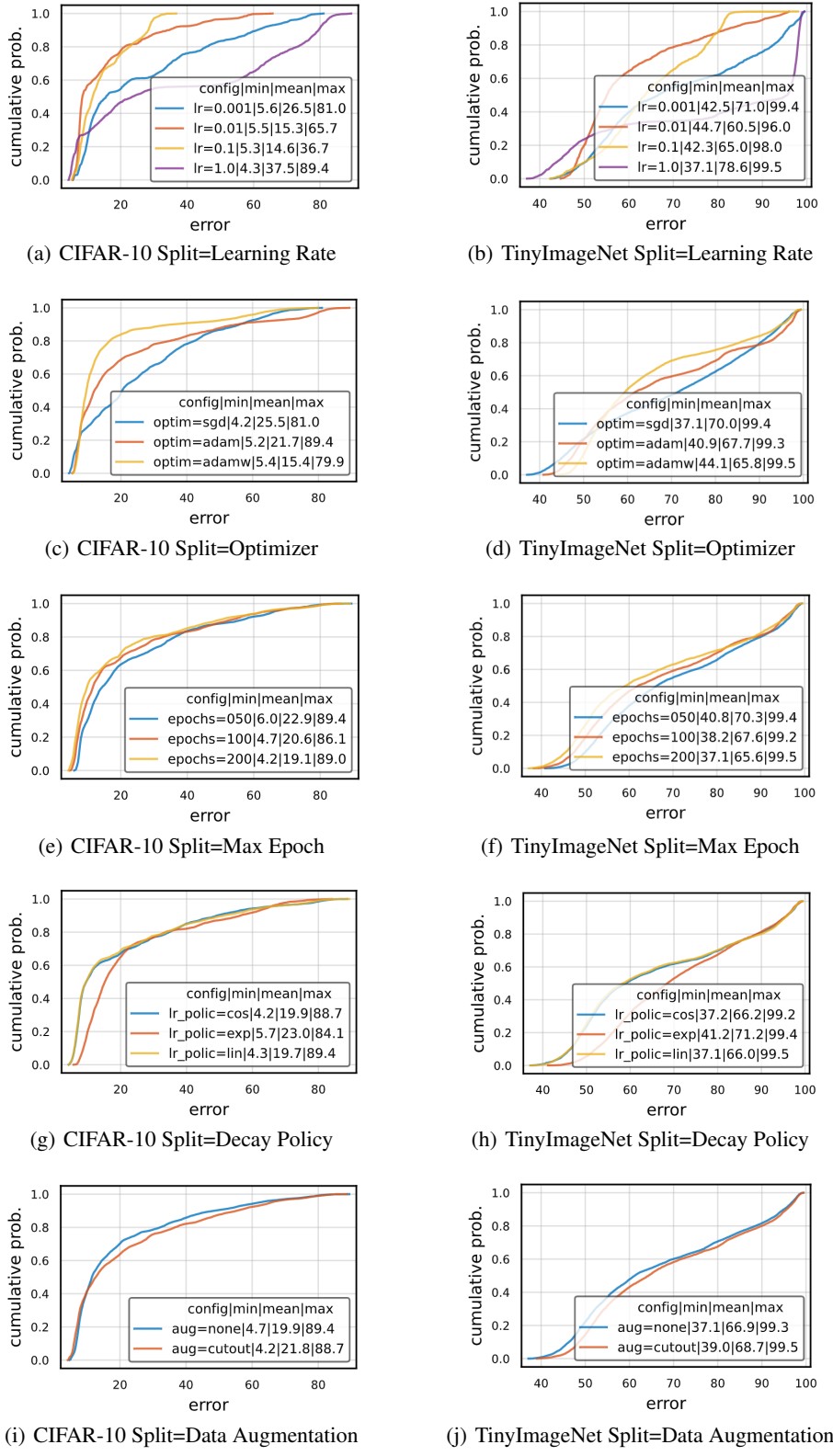

Figure 13: The empirical cumulative distribution (ECDF) of all real measured configurations on CIFAR-10 and TinyImageNet for 5 different splits.

