# OpenReview forum: "EA-HAS-Bench: Energy-aware Hyperparameter and Architecture Search Benchmark"
_ICLR.cc/2023/Conference — ICLR 2023 notable top 25%_

### Official Review · Reviewer_bERE · 2022-10-25

**Confidence:** 4
**Correctness:** 3
**Technical Novelty And Significance:** 2
**Empirical Novelty And Significance:** 3
**Recommendation:** 6

**Clarity, Quality, Novelty And Reproducibility:**

The paper is largely clearly written, with comprehensive experiments for validating the new benchmark dataset. The second objective of demonstrating the usefulness of energy as a conflicting objective to obtain efficient architectures is brief and lacking several details as listed in my main review.


**Strength And Weaknesses:**

**Strengths:**
* The main contribution of this work is the EA-HAS-Bench tabular benchmarking dataset which not only has conventional performance measures, but in addition tabulates training- and inference energy costs. This can become an important contribution as most existing tabular benchmarks mainly report number of parameters or training time as the proxies for efficiency.

* The scale of the new surrogate dataset presented in this work is massive. $6\times 10^7$ unique CNN architectures and 540 hyperparameter configurations are explored yielding about $3\times 10^{10}$ datapoints.

* A novel surrogate model that approximates the learning characteristics using Bezier Curve-based surrogate (BCS) model is presented and validated. Energy and training times are predicted using Surrogate NAS Benchmark approach in [3].

**Weaknesses:**

* **Energy measurements**: The main contribution in this work is on updating the energy consumption as an additional metric. However, the details on how this energy was measured is missing. Which of the components were monitored and on which hardware? It is mentioned that GPU costs are measured; while this is one of the large contributors the CPU, DRAM costs are not small [1]. Factors such as Power usage effectiveness further have impact on the energy efficiency of the models.

* **Surrogate Energy measurements**: As with other metrics, I assume the energy consumption is also estimated using some surrogate methods. In Sec. 2.2 the authors describe that energy costs are measured for one epoch and aggregated over total number of epochs. Authors do mention that energy costs were predicted according to [3] but in [3] there was no energy costs reported. Is this a linear scaling of the energy cost of one epoch? Does this extrapolation extend simply linearly?

* **Updated objective**: The objective that integrates energy consumption with performance is highly heuristic, and presented without any motivation or justification. Given that the key contribution in this work is to use energy consumption as an additional constraint when performing NAS, the specific formulation of the composite objective is quite vague. It is mentioned to be $ACC\times {\frac{TEC}{T}}^w$ with fixed values for T and w. Are there any other ways of incorporating these measurements that were tried? Would the choice of scaling influence the solutions obtained. Do the gains achieved still hold? Why is this specific formulation the preferred choice?

* **Efficiency in terms of FLOPS**: In Sec.4, the correlation (or the lack thereof) between training energy and training times is clearly captured. This is an important result to show the additional resource costs that are missed when only using training time. However, the authors do not extend this analysis or at least discuss how FLOPs as a measure of efficiency correlates (or not) to measuring energy. For instance, works such as [2] use FLOPs as a measure of efficiency and obtain the now famous efficient net architectures.

* **Missing details**: Although the proposed benchmark is a surrogate one, there are no clear reportings of the total cost of obtaining this dataset. For instance, how many architectures/hyperparameter configurations were sampled in the first place to train the two neural networks. This information was either missing or difficult to obtain from the main paper. In Sec 2.4 there's a mention of using random sampling of the search space -- but how many architectures from the $3\times 10^{10}$ architectures were sampled to be approximated using the BCS model? Further, what was the total energy cost/ GPU costs for obtaining this new dataset?

[1] Henderson, Peter, et al. "Towards the systematic reporting of the energy and carbon footprints of machine learning." Journal of Machine Learning Research 21.248 (2020): 1-43.

[2] Tan, Mingxing, and Quoc Le. "Efficientnet: Rethinking model scaling for convolutional neural networks." International conference on machine learning. PMLR, 2019.

[3] Zela, Arber, et al. "Surrogate NAS benchmarks: Going beyond the limited search spaces of tabular NAS benchmarks." Tenth International Conference on Learning Representations. OpenReview. net, 2022.


**Summary Of The Paper:**

This work presents a surrogate tabular benchmark for neural architecture search (NAS) -- EA-HAS-Bench -- with two primary contributions: 1. Joint space comprising architectures and hyperparameter configurations, and 2. Additional metrics to measure efficiency that include training and inference energy costs. This work then uses the EA-HAS-Bench dataset to demonstrate the use of different NAS strategies when constraining the overall budget to a fixed energy/power consumption criterion instead of max. epochs. The work also demonstrates the use of energy consumption per model cost as a factor in the model selection criterion to obtain more energy efficient architectures. Comprehensive experiments are carried out on a large joint space of architectures showing the usefulness of the proposed tabular benchmark dataset.


**Summary Of The Review:**

A new surrogate tabular benchmark which also includes training and inference energy costs, along with other standard performance measures. The dataset itself is large, comprehensive and well designed. The experiments pertaining to surrogate model predictions are reasonably done. The main contribution -- which is the use of energy consumption to design energy efficient architectures is not evaluated rigorously. For instance, the joint objective proposed to integrate energy costs with accuracy is a heuristic with fixed parameters presented without any justification; the impact of using such an objective on the class of obtained architectures is also not clearly investigated. For a tabular benchmark aiming to demonstrate energy efficiency the experiments about this aspect are not strong enough.

---

> ### Author Response · Authors · 2022-11-18
> **Response to Reviewer bERE (Part 3/3)**
>
> ## **Efficiency in terms of FLOPS**
> > **"...However, the authors do not extend this analysis or at least discuss how FLOPs as a measure of efficiency correlates (or not) to measuring energy..."**
>
> We thank the reviewer for this very helpful suggestion. Following the reviewer's suggestion, we compute the Pearson correlation between energy consumption and FLOPs of the models on our dataset and also obtain a relatively low correlation between them (Pearson correlation around 0.5 on TinyImageNet). We add Figure 10 in the Appendix showing the correlation between FLOPs and training energy cost, which also verifies that high FLOPs do not always lead to high energy costs.
>
> This paper does not discuss the energy-FLOPs correlation in detail because this is a problem that has been widely studied in both NAS [1] and the system community [4]. For example, the relationship between FLOPs and energy has been extensively analyzed by HW-NAS-Bench [1]. Table 3-4 of HW-NAS-Bench show that the Kendall Rank Correlation Coefficient of FLOPs and energy is less than 0.5 on multiple datasets and multiple search spaces. HW-NAS-Bench concluded that commonly used theoretical hardware-cost metrics (i.e., FLOPs and #Params) do not always correlate well with measured/estimated hardware cost for the architectures in two search spaces.
>
>
>
> ## **Missing details**
> > **"How many architectures/hyperparameter configurations were sampled in the first place to train the two neural networks? What was the total energy cost/ GPU costs for obtaining this new dataset?"**
>
> We agree with the reviewer that this number is important and it is already reported in "Section 2.5 Surrogate Benchmark Evaluation" of the original paper, *"The sampled configurations on CIFAR10 and TinyImageNet are split into training, validation and testing sets containing 11597, 1288, and 1000 samples respectively."*
>
> For the total energy cost, according to the suggestion of Reviewer QejS and Reviewer bERE, we reported the total energy consumption to build EA-HAS-Bench in Table 9 of the revised paper. We show Table 9 of the revised paper in the following:
>
> Table: The energy consumption (kWh) to build EA-HAS-Bench
> | Dataset      | Training & Validation & Testing   sets | GT(1 seed) | Total   |
> |--------------|-----------------------------------------|--------------|---------|
> | CIFAR10      | 660,313                                  | 46,813        | 707,126  |
> | TinyImageNet | 1,715,985                                 | 124,088       | 1,840,074 |
> | Total        |                                         |              | 2,547,200 |
>
> Table 9 shows the total energy consumption to construct the proposed dataset. It costs about 2.5 million kWh of energy for training around 27,000 deep models in total. On one hand, it shows that HPO/NAS research does require the consumption of large amounts of energy. On the other hand, it also demonstrates that using our dataset instead of actually training these configurations can allow future studies to save a lot of energy. With more and more work using our dataset, the more cost saving of building the dataset will be achieved.
>
> ## **References**
>
> 1. Chaojian Li, Zhongzhi Yu, Yonggan Fu, Yongan Zhang, Yang Zhao, Haoran You, Qixuan Yu, Yue Wang, Cong Hao, and Yingyan Lin. HW-NAS-bench: Hardware-aware neural architecture search benchmark. In 9th International Conference on Learning Representations, ICLR 2021, Virtual Event, Austria, May 3-7, 2021
> 2. Miguel F. Astudillo and Hessam AzariJafari. Estimating the global warming emissions of the LCAXVII conference: connecting flights matter. The International Journal of Life Cycle Assessment, 23(7):1512–1516, Jul 2018. ISSN 1614-7502
>
> 2. Tan M, Chen B, Pang R, et al. Mnasnet: Platform-aware neural architecture search for mobile[C]//Proceedings of the IEEE/CVF Conference on Computer Vision and Pattern Recognition. 2019: 2820-2828.
>
> 3. Bender G, Liu H, Chen B, et al. Can weight sharing outperform random architecture search? an investigation with tunas[C]//Proceedings of the IEEE/CVF Conference on Computer Vision and Pattern Recognition. 2020: 14323-14332.
>
> 4. Anzt H, Haugen B, Kurzak J, et al. Experiences in autotuning matrix multiplication for energy minimization on GPUs[J]. Concurrency and Computation: Practice and Experience, 2015, 27(17): 5096-5113.

---

> > ### Comment · Reviewer_bERE · 2022-12-12
> > **Response to author response**
> >
> > I thank the authors for their thorough and clear responses to most of the concerns raised in my reviews.
> >
> > **Energy consumption measurement**: The details provided in the appendix B.2 is detailed and appears that the authors implemented their own GPU measurement utility. I can point to at least three tools that can do this reliably and report GPU, CPU and DRAM power consumption. [1,2,3] Perhaps these tools might be useful for future work. And according to [1,2] CPU cost is not negligible as it can be around 20%, whereas DRAM costs are about 5% on standard hardware.
> >
> > **Linear scaling from single epochs**: I also agree with the authors that one could reliably make extrapolations from single epochs. However, I would use more than one epoch as there are additional costs of setting up the model, data etc. which are not repeated for subsequent epochs. These might also be more towards the CPU/DRAM costs which this work ignores.
> >
> > **Total energy costs**: I appreciate the reporting of the total energy costs in Table 9. While I agree with the authors that this could be seen as necessary costs and is regularly used as a justification by us (in the NAS community), it still is bothersome. I sincerely hope we can move away from these massively energy intensive experiments.
> >
> > All this said, I am happy with the additional details/ clarifications about objective function, energy measurements, models trained etc. and will raise my score to 8 (accept).
> >
> > **References**:
> > [1] Henderson, Peter, et al. "Towards the systematic reporting of the energy and carbon footprints of machine learning." Journal of Machine Learning Research 21.248 (2020): 1-43.
> >
> > [2] LFW Anthony, B Kanding, R Selvan. Carbontracker: Tracking and Predicting the Carbon Footprint of Training Deep Learning Models
> > ICML Workshop on Challenges in Deploying and monitoring Machine Learning Systems
> >
> > [3] Codecarbon: https://github.com/mlco2/codecarbon

---

> > > ### Author Response · Authors · 2022-12-13
> > > **Response to Reviewer bERE response**
> > >
> > > We thank the reviewer for the new helpful suggestion and raising the score. Here we also provide some further explanations for your concerns.
> > >
> > > ### Energy consumption measurement
> > > > **The details provided in the appendix B.2 is detailed and appears that the authors implemented their own GPU measurement utility. I can point to at least three tools that can do this reliably and report GPU, CPU and DRAM power consumption. Perhaps these tools might be useful for future work.**
> > >
> > > In our paper we implemented our own energy measurement utility, and we agree with reviewer that it will be better to use different tools to measure CPU/DRAM energy consumption and check their consistency. In the future, the energy consumption measured by other tools (as suggested by reviewer) will be added to the proposed dataset.
> > >
> > > ### Linear scaling from single epochs
> > > > **However, I would use more than one epoch as there are additional costs of setting up the model, data etc. which are not repeated for subsequent epochs.**
> > >
> > > We agree with reviewer's point on the energy cost overhead and the overall energy cost should be measured on more than one epoch. In fact, in our experiments, we did measure multiple epochs and excluded the results of the first epoch. The energy consumption of one epoch is the mean consumption of multiple epochs.
> > >
> > > ### Total energy costs
> > > > **While I agree with the authors that this could be seen as necessary costs and is regularly used as a justification by us (in the NAS community), it still is bothersome. I sincerely hope we can move away from these massively energy intensive experiments.**
> > >
> > > We agree with the reviewer that the energy intensive experiments should be avoided. With the obtained dataset, our future research on efficient NAS/HPO algorithms do not require significant energy costs anymore.

---

> ### Author Response · Authors · 2022-11-18
> **Response to Reviewer bERE (Part 2/3)**
>
> ## **Updated objective**
> > **"Are there any other ways of incorporating these measurements that were tried? Would the choice of scaling influence the solutions obtained. Do the gains achieved still hold? Why is this specific formulation the preferred choice?"**
>
>
> **Choice of the objective.** This objective is motivated by previous works on training a model that seeks a better trade-off between accuracy and latency. Specifically, we follow the soft exponential reward function (Soft Exp) in MnasNet [3]. Following the reviewer's suggestion, we compare the soft exponential reward function with some other objective choices, including the hard exponential reward function (Hard Exp), and absolute reward function (Absolute)[4].
>
> Table: The total TEC in reaching target performance on CIFAR10
> | Algorithms    | Origin | Soft Exp | Hard Exp | Absolute |
> |---------------|--------|----------|----------|----------|
> | Local Search  | 5521   | 3218     | 3595     | 6070     |
> | BANANAS       | 4966   | 3630     | 4005     | 5227     |
>
> More detail is shown in Appendix D.4
>
> **Choice of scaling.**  Following MnasNet, we simply set the $\beta$ to -0.07. Following the reviewer's suggestion, we evaluate the performance difference by using different scaling factors $T_0$. Based on the TEC distribution of sampling points mainly between 0.4 and 0.5, we try three different parameters. The results are shown in the table. After $T_0 \geq 0.45$, the final Total TEC is the same, which indicates that EA-NAS is robust to the parameter $T_0$.
>
> Table 10:The impact of different $T_0$ in CIFAR10
> | Algorithms | T =0.4 | T=0.45 | T=0.5 |
> |------------|--------|--------|-------|
> | EA-BNANAS  | 3692   | 3630   | 3630  |
> | EA-LS      | 4751   | 3218   | 3218  |
>
> **Main contribution of the paper.**  We find the reviewer's suggestion on the objective function very insightful and we have added the new ablation study to the revised paper. However, we want to clarify that designing a good energy-aware NAS/HPO algorithm is not the main focus of this paper. Instead, this paper provides a benchmark dataset to accelerate energy-aware NAS/HPO research. In section 5, we only propose a very simple and intuitive way to modify the HPO method to be aware of energy, and we use this simple yet effective baseline to showcase that considering energy cost during architecture and hyper-parameter search is necessary. We believe that energy-aware NAS/HPO is an important open problem and lots of work is waiting to be done to develop much more advanced solutions, and this paper only provides a tool to push this direction forward.

---

> ### Author Response · Authors · 2022-11-18
> **Response to Reviewer bERE (Part 1/3)**
>
> Thank you for your careful review of our work and excellent feedback! Below we try to address all your concerns by providing point-by-point replies.
>
> ## **Energy Measurement**
> > **"The details on how this energy was measured is missing...."**
>
> Based on the reviewer's suggestion, we elaborate on the details of data collection in the “Details of Data Collection” section in Appendix B.2. Specifically, we provide the code on how to measure the energy cost and the details of the machines used to collect energy consumption related information.
>
> > **"Which of the components were monitored and on which hardware?"**
>
> Following energy related metrics are monitored in the proposed dataset:
> - *GPU energy consumption*: Recorded with nvidia-smi on Nvidia Tesla V100 with 32 GB memory
> - *CPU utilization over time*: Recorded on Intel(R) Xeon(R) CPU E5-2690 v3 @ 2.60GHz 2600 MHz. It can be used to estimate the CPU energy cost given the process power of the CPU.
>
> Please note that, based on the observation from training more than 27,000 deep models, we found that compared to GPU energy consumption, the CPU utilization is very low, even negligible. For example, we use a CPU of Intel Xeon E5-2690 v3 @ 2.60GHz with 135 W. For the CIFAR10 dataset, we observe that its utilization is basically around 1\%, and close to 0 for 1/3 of the time. Statistically, the cost of the CPU is only about 2\% of GPU for the whole training process. The energy cost of DRAM is even smaller than CPU, it has a power of 3~5 watts which is less than 5% of the maximum power of CPU. As a result, following many previous works [1][2], in the main paper, we mostly focus on discussing GPU energy consumption. However, with our dataset, one can easily infer CPU energy consumption from CPU utilization over time.
>
> ## **Surrogate Energy measurements**
> > **"I assume the energy consumption is also estimated using some surrogate methods ... Is this a linear scaling of the energy cost of one epoch?"**
>
> The reviewer is correct and the total search energy cost is a linear scaling of the energy cost of one epoch. Here is a detailed introduction on how to estimate the energy cost in this paper.
>
> Following (Ke et al., 2017), LGB is used to predict both runtime and energy cost, were given a training configuration, the model takes the hyper-parameters and architecture parameters as input, and outputs total training time, as well as the average energy cost per epoch.
>
> Specifically, we denote a training configuration in the EA-HAS-Bench search space as $\mathbf{c} \in N^{d}$, where $\mathbf{c}$ is a $d$-dimentional vector containing $d$ training parameters. $e_{sur}(\mathbf{c})$ is the energy cost surrogate model that predicts the training energy cost to train a model with training configuration $\mathbf{c}$ for one epoch. $A = \{\mathbf{c}^{(i)}\}_{i=0}^N$ is the set of configurations a NAS/HPO search method traversed. As a result, the total search energy $e_s$ cost is defined as:
>
> $e_s(A) = \sum_{\mathbf{c}^{(i)} \in A} e_{sur}(\mathbf{c}^{(i)}) * \mathbf{c}^{(i)}_{n}$
>
> where $n$ is the index of $\mathbf{c}$ that stores the number of total training epochs to train the deep model under configuration $\mathbf{c}$. The detailed definition of the search energy cost as well as how the surrogate model is implemented are added to the Appendix B.3 of the revised paper.
>
> > **"Does this extrapolation extend simply linearly?"**
>
> This extrapolation extends linearly because intuitively the operations executed in every epoch are almost identical, and hence the energy consumption of each epoch should be similar. Empirically, we have measured the energy cost of training configurations for several epochs, and we found that the standard deviation of the energy cost per epoch is very low (0.008 kWh on Cifar100 and 0.09 kWh on TinyImageNet). As a result, a linear scale of the energy of one epoch should be a pretty accurate approximation for the total search energy cost.

---

### Official Review · Reviewer_uqiJ · 2022-10-25

**Confidence:** 5
**Correctness:** 4
**Technical Novelty And Significance:** 3
**Empirical Novelty And Significance:** 3
**Recommendation:** 6

**Clarity, Quality, Novelty And Reproducibility:**

The paper is easy to follow and well-written. The authors provide a valid motivation for proposing the new benchmark in the introduction and cover the related work properly. The main novelty aspect of this submission lie in (1) the energy-consumption benchmark that also includes the training cost (differently from HW-NAS-Bench which only has for inference). (2) new surrogate model for learning curve prediction. (3) Joint NAS and HPO search space where the architecture space is the one from RegNet.

In terms of reproducibility the authors do not provide the necessary code together with its API for the benchmark.

**Strength And Weaknesses:**

Below I mention some pros and cons of this submission:

(+) Provides novelty in the field of NAS benchmarking by creating a new surrogate benchmark that includes the AutoML search energy consumption.

(+) Well-written and mainly easy to read.

(+) The surrogate model proposed seems to outperform most of previous surrogate models used in the literature.

(+) Claims supported by empirical evidence and analysis.


(-) I wasn't able to find the codebase for the benchmark, which is one of the most important features when proposing a new NAS benchmark

(-) No description of the API on how to use the benchmark.

(-) Does not support one-shot NAS algorithms.

(-) Section 5 could benefit of more empirical evaluations, as for instance evaluating multi-objective NAS algorithms on the benchmark.

**Questions and other comments**
- I guess the models used to encode the architecture and hyperparameters are MLPs? I could not find more details on them. Did the authors provide these in the text? Also, did they try other surrogate models such as GINs for the architecture?
- It would be interesting to compare the learning curves in Fig 6 to the same counterparts ran on the real benchmark, i.e. without the surrogate predictions, as done in the Surr-NAS-Bench paper.

Minor:
- Wrong reference to NAS-Bench-301 in Section 3.


**Summary Of The Paper:**

This paper proposes EA-HAS-Bench, which is a surrogate neural architecture search and hyperparameter benchmark that can be used to query the energy cost (in kWh) of training and inference of neural networks in the search space. This can be used as another metric when benchmarking AutoML algorithm to compare them with the energy consumption, which according to the empirical results in the paper, is not well aligned with the training costs. The authors create the benchmark on 2 popular image classification datasets, namely CIFAR-10 and Tiny-ImageNet.

**Summary Of The Review:**

In general the paper provides a useful contribution to the NAS community, by proposing a new benchmark that provides the energy consumption utilized to train and evaluate a configuration. Even though suited mainly for black-box algorithms the search space is useful and the surrogate model used to predict the learning curves is novel in the NAS literature. The claims are backed with empirical evidence. I lean towards acceptance for this paper.

---

> ### Author Response · Authors · 2022-11-18
> **Response to Reviewer uqiJ (Part 2/2)**
>
> ## **Other surrogate models such as GINs for the architecture**
> > **"I guess the models used to encode the architecture and hyperparameters are MLPs? I could not find more details on them. Did the authors provide these in the text? Also, did they try other surrogate models such as GINs for the architecture?"**
>
> You are right and the encoders for the architecture are MLPs. Specifically, two MLPs are adopted as network architecture and hyper-parameter encoder, respectively. Each MLP contains two linear transformation layers with a ReLU activation function. Then, the encoded features are fed into the learning curve prediction network, which is also an MLP with an extra dropout layer, whose output is fed into two linear regressors that output the coordinates of the control points. We have added the details of the BSC model in Appendix B.4.
>
> Furthermore, as shown in Table 3 we did try other surrogate models, such as tree-based methods like LGB or XGB. We thank the reviewer's suggestion to adopt GIN for the architecture encoder. However, GIN might not be feasible in our proposed search space. On one hand, different from works like NAS-Bench-101 and 201 which focus on network connections, our architecture space focuses on network shapes (e.g., depth, width, slope, etc), which is difficult to be modeled by the graph-based method. On the other hand, our search space also contains hyperparameters, which also do not have a strong topological connection suitable for graph learning.
>
> ## **Experiments on the real benchmark**
> > **"It would be interesting to compare the learning curves in Fig 6 to the same counterparts ran on the real benchmark, i.e. without the surrogate predictions, as done in the Surr-NAS-Bench paper."**
>
> Following the Reviewer's suggestion, we conduct the experiments to run methods in Fig 6 on a real  dataset. Considering that some NAS algorithms such as Local Search cannot be applied in a space of only sampled points, we implement these NAS algorithms on a newly constructed small **tabular** benchmark of size 500. We have added a detailed introduction and analysis of this small real benchmark into the appendix E.
>
> In this experiment, the maximum search cost is set to 20,000 kWh, which is equivalent to running a single-fidelity HPO algorithm for about 200 iterations. We run 10 trials of each AutoML algorithm and compute the mean and standard deviation. The result is shown in Figure of Appendix E. Due to the different search spaces and budgets, the conclusions drawn are a bit different from the previous ones on the surrogate benchmark.
>
>
> ## **Wrong reference to NAS-Bench-301 in Section 3**
> Thanks to the reviewer for pointing out the error. We have fixed it by replacing it with the right reference.
>
> ## **Reference**
> 1. Tan M, Chen B, Pang R, et al. Mnasnet: Platform-aware neural architecture search for mobile[C]//Proceedings of the IEEE/CVF Conference on Computer Vision and Pattern Recognition. 2019: 2820-2828.
> 2. Bender G, Liu H, Chen B, et al. Can weight sharing outperform random architecture search? an investigation with tunas[C]//Proceedings of the IEEE/CVF Conference on Computer Vision and Pattern Recognition. 2020: 14323-14332.

---

> ### Author Response · Authors · 2022-11-18
> **Rsponse to Reviewer uqiJ(Part 1/2)**
>
> Many thanks for your positive feedback and the acceptance score. We provide the point-by-point replies in the following:
>
> ## **Codebase for the benchmark**
> > **"I wasn't able to find the codebase for the benchmark, which is one of the most important features when proposing a new NAS benchmark"**
>
> Please refer to the "Code Release" thread where we put our code base in an anonymous link that only ACs and reviewers can see during the paper reviewing phase.
>
> ## **The API on how to use the benchmark**
>
> According to the reviewer's suggestion, we have added the API of EA-HAS-Bench in Appendix B.6.
>
>
> ## **Not support one-shot NAS algorithms**
>
> To consider more factors affecting energy cost, besides architecture, our search space also contains training hyper-parameters. However, it is a known limitation of existing one-shot NAS algorithms, that they are only suitable for search space containing only architecture. As a result, we agree with the reviewer that this method mainly focuses on HPO and non-weight-sharing NAS methods. We think it is a great suggestion to develop a benchmark dataset for one-shot NAS methods and we will explore this direction in our future work.
>
> ## **More Empirical Evaluations**
> > **"Section 5 could benefit of more empirical evaluations, as for instance evaluating multi-objective NAS algorithms on the benchmark."**
>
> According to the Reviewer's suggestion, we conduct a more comprehensive empirical evaluation of the proposed new energy-aware baselines. Specifically, an ablation study is conducted to examine how different reward functions and hyper-parameters in the proposed objective affect the performance of the energy-aware HPO, including reward functions like soft exponential reward function (Soft Exp), hard exponential reward function (Hard Exp) [1] and absolute reward function (Absolute) [2]. These existing comparison methods  transform the multi-objective optimization problem into a single-objective optimization problem and design a suitable reward function so that the optimal solution of the single-objective optimization problem is an efficient solution/weakly efficient solution of the multi-objective optimization problem.
> The following table shows the total energy cost for the compared methods to achieve target performance. A detailed introduction is added to Appendix D.4 of the revised paper.
>
> Table: The total TEC in reaching target performance on CIFAR10
> | Algorithms    | Origin | Soft Exp | Hard Exp | Absolute |
> |---------------|--------|----------|----------|----------|
> | Local Search  | 5521   | 3218     | 3595     | 6070     |
> |  BANANAS       | 4966   | 3630     | 4005     | 5227     |

---

### Official Review · Reviewer_ZDyf · 2022-10-30

**Confidence:** 3
**Correctness:** 4
**Technical Novelty And Significance:** 3
**Empirical Novelty And Significance:** 3
**Recommendation:** 8

**Clarity, Quality, Novelty And Reproducibility:**

Quality:

Important point: "provides the full training information including training and test accuracy learning curves" What about validation curves? Validation curves are typically used for early stopping & choosing between models. Test error should only be used at the end of a search, not during the search (NAS-Bench-101, Ying et al., 2019). Hopefully this is just a typo.

Novelty:

This paper seems novel to me. It is of course building on a series of previous papers, but adds several contributions. To me, the main one is having a dataset for NAS-HPO reporting energy cost. However, other contributions, like providing a large search space for NAS-HPO joint search and using a new surrogate method that enables longer learning curve prediction, are also valuable, even for researchers not trying to balance energy cost. (However, I hope that balancing accuracy & energy cost becomes more common.)

Reproducibility:

The authors plan to share the dataset they created, which will enable reproducible HAS-NAS studies. However, it would be helpful to also share the code. If many people use the dataset, it will be important to fully understand how it was created. Also, the BSC surrogate model is fairly complicated, and future research would greatly benefit from being able to use the code instead of trying to recreate it. It is helpful that the hyperparameters of the surrogate models (both for predicting the learning curves & for predicting the other metrics) are listed in the appendix, though.

"For different datasets (i.e., CIFAR-10 and TinyImageNet), we adjust this space accordingly via constraining parameters and FLOPs." Is this ever explained?

Figure 6 contains acronyms for a variety of NAS algorithms that I don't believe are ever explained. For example, what is REA? Using Appendix D I can guess what the acronyms mean, but I think the acronyms should be explicitly explained, along with citations, in the main paper. Also, there is not much detail for this figure (Section 5.1 & Appendix D), so it would be difficult to reproduce.

Clarity:

"LGB models" are introduced on page 4 but the acronym is never explained. From Googling, I'm guessing this is Light Gradient Boosting?

"The energy cost surrogate models achieve 0.787 for R2 and 0.686 for KT on CIFAR-10 and 0.959 for R2 and 0.872 for KT on TinyImageNet." How do these numbers compare to previous papers that use surrogate models to fit a search space, such as NAS-Bench-301, NAS-Bench-X11, and NAS-HPO-Bench-II? I don't have a sense for if these metrics are good enough to make the surrogate useful for NAS/HPO research.

What is GT in Table 3 & Figure 8? In Figure 8, it seems to mean "ground truth," but I'm not sure what "ground truth (1 seed)" would mean in Table 3.

The fonts in the figures are often hard to read.

There's something missing here: "we new energy-aware AutoML method that arms"

"Our dataset needs to provide the total search energy cost of running a specific AutoML method, which is the sum of the training energy cost of all the hyperparameter and architecture configurations the search method traverses." Technically, the dataset provides the energy cost of particular configurations, which then enables someone trying an AutoML method to calculate the total search energy cost, correct?

"The size of the maximum number of epochs is almost proportional to the quality of the search space..." I don't understand this sentence, but perhaps I don't fully understand how to associate the ECDF with search space quality. For the other hyperparameters, I can clearly see in Figure 3 that changing the hyperparameters changes the distribution.

The paper writing is fairly verbose, often repeating the same points. If you need to fit in more material to address reviews, I think you could condense some of the existing writing. I especially noticed the repetitiveness once I got to page 4.

On page 2, "EE-HAS-Bench" is used instead of "EA-HAS-Bench."

"Tree-of-Parzen-Estimators (TPE) (Bergstra et al., 2011) is adopted for BCS to search for the best hyperparameter configuration." and "For the optimal three surrogate models in Table 3 and the LGB energy cost model (LGB-E), the optimal parameters found using TPE and the search space are listed in Table 5." So only the best surrogate models get their hyperparameters tuned? Perhaps the other surrogate models appear inaccurate by bad luck?

"Since the prediction results may have anomalies that make the prediction performance much higher than the actual results, we introduce spike anomalies (Yan et al., 2021) to evaluate these anomalies." This is vague, and I don't know what it means. Spike anomalies are mentioned again in the next paragraph, where I also don't understand them.

"On the other hand, the importance of the different parameters varies considerably, especially exponentially." I'm not sure what this means, especially the "especially exponentially" part.

In the middle plot of Figure 4, I don't know what the orange means. I also thought it was unclear how we can tell that models in the Pareto Frontier on the right are not always on the Pareto Frontier in the middle figure, so perhaps that's what orange is for?

Figure 5 would be easier to read with a diverging color palette with 0 being white.

The description of Figure 5 in its caption and in the text would suggest that you're only showing correlation coefficients for accuracy & TEC. However, there are 4 columns.

"Energy-aware BANANAS and vs. origin BANANAS and LS" I'm guessing this is supposed to be "Energy-aware BANANAS and Energy-aware LS vs. origin BANANAS and LS"

In Figure 6 (right), I'm guessing that the y-axis should be accuracy? If so, this figure and its description Section 5.2 tell a clear story that modifying these HPO algorithms to consider energy cost results in being able to find good solutions with less energy. However, the story is less clear for Figure 6 (left) and Figure 6 (middle). I see that with a limited energy budget, LS and bananas reach the lowest regret for CIFAR10, and BOHB and HB reach the lowest regret for TinyImageNet. Are these different conclusions than if energy was not considered?

**Strength And Weaknesses:**

I agree that this is an important problem, and I think that this dataset could be very helpful for saving energy during future research. Also, including HPO makes a lot of sense to me, since a lot of energy could be saved by a model that converges faster.

I have various questions and comments below to improve the writing of the paper. I am also concerned about the reproducibility, as discussed below.


**Summary Of The Paper:**

This paper presents a new benchmark dataset for joint neural architecture search (NAS) & hyperparmeter optimization (HPO). It includes information about the energy cost of training a model so that people studying NAS-HPO algorithms can incorporate energy cost. They use a large search space for joint NAS-HPO search and develop a new surrogate model for this search space. In particular, this surrogate model can predict learning curves of arbitrary length. They also demonstrate how to use this dataset.

**Summary Of The Review:**

Overall, I like this paper, although the writing & reproducibility could be improved. I'll tentatively mark this paper as "accept," with the assumption that the authors can make at least some of the points more clear. Also, I'm hoping to confirm that the validation curves are provided in the dataset, not just training & test. If validation curves are not provided, I'll lower the score.

**Update post-rebuttal**

Thanks for improving the clarity and confirming that the validation curves are included. I'll keep the score at "8: accept, good paper."

---

> ### Author Response · Authors · 2022-11-18
> **Response to Reviewer ZDyf (Part 4/4)**
>
> ## **Clarity (Part 3/3)**
>
> ### **What Spike anomalies mean**
>
> Metrics like R2 or KT evaluate the performance of the predicted learning curve based on the overall statistics and are not sensitive to anomalies. However, very few anomalies on the learning curve can already affect the final performance prediction. Hence, following NAS-Bench-X11 [1], we use spike anomalies to evaluate how often the anomalies appear on the curve. Considering the space limitation, we refer the reader to the original paper of NAS-Bench-X11 for a detailed introduction to spike anomalies. We agree with the reviewer that this may cause confusion to the users. According to the reviewer's comments, we also give a clearer explanation of why we use the spike anomalies in section 2.5. Furthermore, we have included a detailed description of spike anomalies in the Appendix. Specific contexts are as follows:
>
> *"Details of Spike Anomalies. Although  R2 and KT can evaluate the surrogate model by measuring the overall statistics between the surrogate benchmark and the ground truth, they are not sensitive to anomalies. Following NAS-bench-X11 \cite{nas-bench-x11}, to evaluate the performance of surrogate models based on anomalies, we introduce the Spike Anomalies metrics. We first calculate the largest value $x$ such that there are fewer than 5\% of learning curves whose maximum validation accuracy is higher than their final validation accuracy, on the true learning curves. Next, the percentage of surrogate learning curves whose maximum validation accuracy is $x$ higher than their final validation accuracy was computed."*
>
> ### **What "especially exponentially" means**
>
> The previous writing is not clear enough. What we want to express is that the importance of different parameters in a surrogate model is different. For example, for $a$, $c$, and $\alpha$ in pow$_3$ ($c-ax^{-\alpha}$), the $\alpha$ in exponent has a higher impact on the overall learning curve. The revised sentence is as follows：
>
> *"... the importance of the different parameters in a surrogate model varies considerably, especially the parameter which is in the exponent of an exponential function. "*
>
> ### **orange cross in Figure 4**
> > **"In the middle plot of Figure 4, I don't know what the orange means. I also thought it was unclear how we can tell that models in the Pareto Frontier on the right are not always on the Pareto Frontier in the middle figure, so perhaps that's what orange is for?"**
>
> The reviewer's understanding is correct. The red dots in both the middle and right plots indicate the points with the best accuracy and runtime trade-off (i.e. points on the Pareto Frontier of the right figure), while the orange cross denotes the red dots that are not on the Pareto Frontier in the middle figure. Based on the reviewer's comments, we added the meaning of the orange cross to the legend in the middle of figure 4, as well as the caption.
>
> ### **The caption and color palette of Figure 5**
>
> According to the reviewer's suggestions, we have modified the caption and color palette of Figure 5.
>
> ### **Energy-aware BANANAS and Energy-aware LS vs. origin BANANAS and LS**
>
> Thanks to the reviewer for pointing this out, we have modified the caption in Figure 5.
>
> ### **Story of Figure 6 (left) and Figure 6 (middle)**
> > **"In Figure 6 (right), I'm guessing that the y-axis should be accuracy? ...  the story is less clear for Figure 6 (left) and Figure 6 (middle)."**
>
>
> Thanks to the reviewer for pointing out the typo in Figure 6. The y-axis should be accuracy and we have fixed the error. We want to clarify that the purpose of Figure 6 (left) and Figure 6 (middle) is mainly to benchmark existing HPO methods and compare their performance on the accuracy/energy cost trade-off on our dataset. This is why we change the search budget to energy cost from the runtime in previous works. Note that, the compared methods in Figure 6 (left) and Figure 6 (middle) do not explicitly consider the energy cost. This is why we propose two new energy-aware baselines in section 5.2 and Figure 6 (right).
>
> ## **References**
>
> [1] Shen Yan, Colin White, Yash Savani, and Frank Hutter. Nas-bench-x11 and the power of learning curves. In Advances in Neural Information Processing Systems 34: Annual Conference on Neural Information Processing Systems, pp. 22534–22549, 2021.

---

> ### Author Response · Authors · 2022-11-18
> **Response to Reviewer ZDyf (Part 3/4)**
>
> ## **Clarity (Part 2/3)**
>
> ### **Meaning of "GT" in Table 3 and Figure 8**
> > **"What is GT in Table 3 & Figure 8? In Figure 8, it seems to mean "ground truth," but I'm not sure what "ground truth (1 seed)" would mean in Table 3."**
>
> ***GT in Table 3:*** "GT" in Table 3 means, we actually train the configurations in the test set for two rounds and the "GT" row of the table evaluates the consistency between the two rounds of training with different random seeds. Our surrogate model achieves similar consistency compared to the consistency between two rounds of actual training, which shows the generated data should be usable. In other words, "GT" is equivalent to comparing our proposed method with a 1-seed tabular benchmark. According to the comments of Reviewer QejS and ZDyf, we add "GT means the 1-seed tabular benchmark" in the caption of Table 3 and clearly point out that we have trained the testing set with 2 seeds and one of the two seeds of the test set is used as the ground truth.
>
> ***GT in Figure 8:*** GT in Figure 8 is the actual learning curve from training the configuration. We add this clarification in the appendix of the revised paper.
>
> ### **Fonts in the figures**
>
> Thanks to the reviewer's comments, we adjusted the font size from 18 to 22 in the figure to make it easier for readers to read.
>
> ### **Something missing**
> > **"There's something missing here: "we new energy-aware AutoML method that arms""**
>
> Thanks to the reviewer's comments, we have refined the writing. The sentence should be "we modify existing AutoML algorithms to consider the search energy consumption"
>
>
> ### **Understanding of the energy cost is correct**
> > **""Our dataset needs to provide the total search energy cost of running a specific AutoML method, which is the sum of the training energy cost of all the hyperparameter and architecture configurations the search method traverses." Technically, the dataset provides the energy cost of particular configurations, which then enables someone trying an AutoML method to calculate the total search energy cost, correct?"**
>
> This understanding is exactly what we want to express. We have modified this expression based on the reviewer's suggestions:
> *"Our dataset needs to provide the total search energy cost of running a specific AutoML method. This can be obtained by measuring the energy cost of each particular configuration the method traverses and summing them up."*
>
> ### **The expression regarding ECDF with different numbers of epochs are confusing**
>
> What we mean here is that the total number of epochs and the model performance are positively correlated, where a larger number of epochs result in better search space (with a greater area under the ECDF curve).
> We agree with the reviewer that this expression could be confusing and we change it from "proportional" to "positively correlated" in the revised paper.
>
>
> ### **Condense some of the existing writing, especially on page 4**
>
> Thanks to this reviewer's suggestion, we delet the duplicate content on page 4.
>
> ### **On page 2, "EE-HAS-Bench" is used instead of "EA-HAS-Bench"**
>
> Thanks to the reviewer for pointing out this error and we have fixed it in the revised paper.
>
> ### **Only the best surrogate models get their hyperparameters tuned?**
> > **""Tree-of-Parzen-Estimators (TPE) (Bergstra et al., 2011) is adopted for BCS to search for the best hyperparameter configuration." and "For the optimal three surrogate models in Table 3 and the LGB energy cost model (LGB-E), the optimal parameters found using TPE and the search space are listed in Table 5." So only the best surrogate models get their hyperparameters tuned? Perhaps the other surrogate models appear inaccurate by bad luck?"**
>
> Our previous description may have caused some misunderstanding. In fact, for a fair comparison, the hyper-parameters of all the surrogate models are tuned in the same way with the validation set.
> In table 5, to save space, we only list the hyperparameters of the best three models, this does not mean the rest of the model's hyperparameters are not tuned. We have given a more clarified description of the hyperparameter tuning in section 2.5 and AppendixB.2 of the revised paper.
>
> Specifically, for the first sentence the reviewer quoted in section 2.5, we revise it as follows：
>
> *"Tree-of-Parzen-Estimators (TPE) is adopted for **all surrogate models** to search for the best hyperparameter configuration."*
>
> The second sentence quoted by the reviewer in AppendixB.2 is revised to the following paragraph:
>
> *"Table 3 shows the optimal hyper-parameters searched by TPE for different surrogate models. Due to the page limit, here we only listed the hyper-parameters of the three models that achieve the best performance in Table 5. "*

---

> ### Author Response · Authors · 2022-11-18
> **Response to Reviewer ZDyf (Part 2/4)**
>
> ## **Clarity (Part 1/3)**
> According to the reviewer's comments, we improve the clarity of the paper. We provide the point-by-point replies in the following:
>
> ### **"LGB models" is never explained**
>
> Thanks to the reviewer, "LGB" denotes "Light Gradient Boosting" (LightGBM, Ke et al., NeurIPS 2017). We have shown the full name of “LBG” where “LGB” first appears and added the related reference.
>
> ### **Performance of energy cost surrogate model**
> > **"How do these numbers compare to previous papers that use surrogate models to fit a search space...I don't have a sense for if these metrics are good enough to make the surrogate useful for NAS/HPO research"**
>
> Since the vast majority of existing benchmarks do not focus on search energy consumption, we cannot compare them with NAS-Bench-301, NAS-Bench-X11, and NAS-HPO-Bench-II. HW-NAS-Bench  focuses on inference hardware consumption, which is similar to ours, and the Pearson correlation coefficients between the approximated and measured Edge GPU energy on are 0.83 on CIFAR100 and 0.93 on ImageNet. In comparison, the Pearson coefficient between our energy cost surrogate model and measured Tesla V100 energy is 0.89 on Cifar10 and 0.97 on TinyImageNet, which shows the surrogate model prediction is quite close to the real energy measurement and hence is good enough for NAS/HPO research. This evaluation result is added to section 2 of the revised paper.

---

> ### Author Response · Authors · 2022-11-18
> **Response to Reviewer ZDyf (Part 1/4)**
>
> Thank you for your thoughtful review. We are glad to hear that you found the paper of good quality and that you think it will contribute to the NAS community. Below we try to address all your concerns.
>
> ## **Validation Curve**
>
> > **“Important point: "provides the full training information including training and test accuracy learning curves" What about validation curves? Validation curves are typically used for early stopping & choosing between models. Test error should only be used at the end of a search, not during the search (NAS-Bench-101, Ying et al., 2019). Hopefully this is just a typo.”**
>
> We forget to mention the validation learning curve in the expression "provides the full training information including training and test accuracy learning curves" in the original paper. We thank the reviewer for pointing out this mistake. Our dataset stores all three curves, i.e. training, validation, and test curve.
>
> In fact, in our paper, some analysis is conducted on the validation curves. For example, in Figure 2 we have used the real measured validation accuracy for comparison with three Benchmarks. Besides training, validation, and test accuracy, we also store the checkpoints of $13885\times2$ models at each epoch (these weights add up to nearly 100TB, which we will release the checkpoints with the dataset.)
>
> ## **Code Release**
> > **" it would be helpful to also share the code"**
>
> Please refer to the "Code Release" thread where we put our code base in an anonymous link only ACs and reviewers can see.
>
>
> ## **Explain following expression**
>
> > **"For different datasets (i.e., CIFAR-10 and TinyImageNet), we adjust this space accordingly via constraining parameters and FLOPs."**
>
> The current writing may have caused some misunderstanding. This expression means that the original search space of $\mathrm{RegNet}$ is designed for ImageNet and is non-trivial to directly apply to CIFAR10 or TinyImageNet. As a result, we adjust the original RegNet space by shrinking down the original model size range (smaller depth, width, etc) of the RegNet space and constraint the total parameters and FLOPs of a model to a relatively small size, which achieves faster training and saves resources. This explanation is added to section 2.1 of the revised PDF.
>
> ## **Acronyms and details of Figure 6 should be explained**
>
> > **“Figure 6 contains acronyms for a variety of NAS algorithms that I don't believe are ever explained.  I think the acronyms should be explicitly explained, along with citations, in the main paper. ”**
>
> Due to space constraints, we originally put the explanation of the acronyms in the appendix, which may confuse the reader. Per the reviewer's suggestion, this explanation is added to Section 5 of the main paper. Specific contexts are as follows:
>
> *"we evaluate the trade-off between search energy cost and model performance of four single-fidelity algorithms: random search (RS) (Li & Talwalkar, 2019), local search (LS) (White et al., 2020), regularized evolution (REA) (Real et al., 2019), BANANA (White et al., 2021a), and two multi-fidelity bandit-based algorithms: Hyperband (HB) (Li et al., 2017) and Bayesian optimization Hyperband (BOHB) Falkner et al. (2018). The implementation details of the above algorithms are in Appendix D. "*
>
> > **"Also, there is not much detail for this figure (Section 5.1 & Appendix D), so it would be difficult to reproduce"**
>
> Following the reviewer's suggestion, we add a more detailed implementation of the experiments in Figure 6 to appendix D. Specific contexts are as follows:
>
> *"Details of the Experimental Implementation. Following Ying et al. (2019) and Yan et al. (2021), during the search, we keep track of the best architectures found by the algorithm after each evaluation and rank them according to their validation accuracy. When the metric we specify (e.g., total energy consumption or target accuracy) exceeds the limit we set, we stop the search. After the search, we query the corresponding best accuracy of the model. We then compute regret:
> $regret_i = Acc_i − Acc^∗$
> where $Acc_i$ denotes the accuracy of the best architecture after each evaluation $i$ and $Acc^∗$ denotes the model with the highest average accuracy in the entire dataset. For experiments in section 5, we run 5 trials of each AutoML algorithm and compute the mean and standard deviation."*
>
> Furthermore, all the codes of the methods in Figure 6 will be publicly released, so that anyone can easily reproduce our results.

---

### Official Review · Reviewer_QejS · 2022-11-01

**Confidence:** 4
**Correctness:** 3
**Technical Novelty And Significance:** 3
**Empirical Novelty And Significance:** 3
**Recommendation:** 6

**Clarity, Quality, Novelty And Reproducibility:**

* This paper is well-written. The contribution is clearly explained.
* Although several NAS and AutoML benchmark datasets already exist, the difference and novelty compared to existing ones are clearly described. EA-HAS-Bench is the first benchmark dataset that considers energy consumption and is a large-scale surrogate-based benchmark of architecture/hyperparameter joint search space.
* The dataset and code are not provided in the current phase, while the authors state that the dataset of EA-HAS-Bench will be released after the paper publication. The reviewer encourages the authors to release the code of the dataset collection and the code of experiments using the dataset.

**Strength And Weaknesses:**

[Strength]
* A novel benchmark dataset for AutoML/NAS is provided, which includes search energy consumption and architecture/hyperparameter joint search space. As the common benchmark datasets are useful to develop and compare the HPO/NAS algorithms and EA-HAS-Bench considers a novel aspect of benchmarking, it will be valuable in the community.
* The demonstration and benchmarking of existing algorithms using EA-HAS-Bench are sound.

[Weaknesses]
* The surrogate model-based benchmark datasets might have a gap with a really collected dataset. It might be better to provide not only the large-scale surrogate dataset but also the small "real" dataset.
* There exist several unclear points. Please see the following comments.

[Comments for Authors]
* The reviewer is not sure of the definition of energy consumption in this paper. How did the authors measure the search energy consumption? The detail of the data collection method, including machine spec, should be reported.
* The reviewer could not find the number of actually evaluated samples to build the surrogate model. Such information should be reported to ensure the reliability of datasets.
* The authors use the Bezier curve and train the model that predicts control points for learning curve prediction. The reviewer thinks that the Bezier curves of order $n$ are equivalent to polynomial functions of order $n$. Therefore, we can use the model that predicts the coefficients of polynomial functions of order $n$ instead of predicting the control points of Bezier curves. The advantage of predicting the control points of Bezier curves should be clarified.
* It would be better to report the total energy consumption to build EA-HAS-Bench.
* What does "GT" mean in Table 3?
* What does "LC" mean in Figure 6?
* Why is the result of BOHB for CIFAR-10 in Figure 6 omitted?
* It would be nice if the authors mentioned the following recent paper. The following paper seems to relate to this work, although it might be published after this paper's submission.

Archit Bansal, Danny Stoll, Maciej Janowski, Arber Zela, Frank Hutter, "JAHS-Bench-201: A Foundation For Research On Joint Architecture And Hyperparameter Search," NeurIPS 2022 Track Datasets and Benchmarks.
https://openreview.net/forum?id=_HLcjaVlqJ


**Summary Of The Paper:**

This paper provides a surrogate model-based benchmark dataset for neural architecture search, which includes search energy consumption and architecture/hyperparameter joint search space. The proposed dataset, called EA-HAS-Bench, enables us to compare the NAS method in energy-aware settings. A Bezier curve-based predictive model is used to provide the surrogate learning curves. The authors exhibit the benchmarking results of existing HPO/NAS algorithms on the energy-aware setting by using the proposed EA-HAS-Bench.

**Summary Of The Review:**

This paper provides a novel HPO/NAS benchmark dataset that is valuable for the community. However, several unclear points should be addressed.

---

> ### Author Response · Authors · 2022-11-18
> **Response to Reviewer QejS (Part 3/3)**
>
> ## **Clarity (Part 2/2)**
>
> ### **Total energy consumption**
> > **“It would be better to report the total energy consumption to build EA-HAS-Bench.”**
>
> According to the suggestion of Reviewer QejS and bERE, we reported the total energy consumption to build EA-HAS-Bench in Table 9 of the revised paper. We show Table 9 in the following:
>
> Table: The energy consumption (kWh) to build EA-HAS-Bench
> | Dataset      | Tranining & Validation & Testing   sets | GT(1 seed) | Total   |
> |--------------|-----------------------------------------|--------------|---------|
> | CIFAR10      | 660,313                                  | 46,813        | 707,126  |
> | TinyImageNet | 1,715,985                                 | 124,088       | 1,840,074 |
> | Total        |                                         |              | 2,547,200 |
>
> Table 4 shows the total energy consumption to construct the proposed dataset. It costs nearly 2.5 million kWh of energy for training around 27,000  deep models in total. On the one hand, it shows that HPO/NAS research does require the consumption of large amounts of energy. On the other hand, it also demonstrates that using our dataset instead of actually training these configurations can allow future studies to save a lot of energy. With more and more work using our dataset, the more cost saving of building the dataset will be achieved.
>
> ### **Meaning of "GT" in Table 3**
> > **“What does "GT" mean in Table 3?”**
>
> "GT" in Table 3 means, we actually train the configurations in the test set for two rounds and the "GT" row of the table compares consistency between the two rounds of training.  Our surrogate model achieves similar consistency compared to the consistency between two rounds of actual training, and the generated data  should be usable. In other words, we can compare our proposed method with a 1-seed tabular benchmark.
>
> According to the comments of Reviewer QejS and ZDyf, we add "GT (1 seed) means the 1-seed tabular benchmark" in the caption of Table 3 and clearly point out that we have trained the testing sets with 2 seeds and one of the two seeds of the test set is used as the ground truth.
>
> ### **Meaning of "LC" in Figure 6**
> We think what the reviewer means should be "LS" in Figure 6. "LS" refers to Local Search and we have refined the caption of Figure 6. Considering the space limitation， we have explained the details of NAS algorithms in Appendix D.
>
> ### **BOHB omitted in CIFAR-10 of Figure 6**
> Thanks for your suggestion, and we have added the BOHB to the left plot of Figure 6 in the revised paper.
>
> On CIFAR-10, HB and BOHB achieved performance similar to that of random search. The possible reason for this is that the correlation between the relative rankings of architectures using low and high fidelity in EA-HAS-Bench CIFAR10 is low (similar to NAS-Bench-101 and NAS-Bench-201) and the HB-based methods will predict the final accuracy of partially trained architectures directly from the final trained accuracy [1].
>
> ### **More related work**
> We thank the reviewer's suggestion and add the JAHS-Bench-201 [2] to related work. JAHS-Bench-201 also has a large-scale joint search space. JAHS-Bench-201 provides FLOPS, Latency, and Runtime in addition to performance and loss. However, JAHS-Bench-201 does not focus on energy consumption during the search.
>
>
> ## **Code Release**
>
> > **“The reviewer encourages the authors to release the code of the dataset collection and the code of experiments using the dataset.”**
>
> We have added the partial code related to how to collect GPU information in the appendix. The entire codebase will be released next month and we now provide an anonymous repo that can only be seen by reviewers and ACs (please refer to the "Code Release" Thread).
>
>
> ## **Reference**
>
> [1] Shen Yan, Colin White, Yash Savani, and Frank Hutter. Nas-bench-x11 and the power of learning curves. In Advances in Neural Information Processing Systems 34: Annual Conference on Neural Information Processing Systems, pp. 22534–22549, 2021.
>
> [2] Archit Bansal, Danny Stoll, Maciej Janowski, Arber Zela, Frank Hutter, JAHS-Bench-201: A Foundation For Research On Joint Architecture And Hyperparameter Search, NeurIPS 2022 Track Datasets and Benchmarks

---

> ### Author Response · Authors · 2022-11-18
> **Response to Reviewer QejS (Part 2/3)**
>
> ## **Clarity (Part 1/2)**
>
> > **"There exist several unclear points"**
>
> We provide the point-by-point replies in the following:
>
>
> ### **Details on search energy consumption**
> > **"How did the authors measure the search energy consumption? Detail of the data collection should be reported.”**
>
> We introduce the definition of search energy consumption in the first paragraph of page 2 and section 2.2 "EVALUATION METRICS". Intuitively, the search energy cost is the total energy consumption to complete a search algorithm. Since the majority of the energy cost comes from training each deep model the search algorithm traverses, in our dataset, the search energy cost is defined as: *"the total energy cost (in kWh) or time (in seconds) to train the model configurations traversed by the search algorithms."*
>
> Specifically, we denote a training configuration in the EA-HAS-Bench search space as $\mathbf{c} \in N^{d}$, where $\mathbf{c}$ is a $d$-dimentional vector containing $d$ training parameters. $e_{ep}(\mathbf{c})$ is the energy cost measure function that returns the training energy cost to train a model with training configuration $\mathbf{c}$ for one epoch. $A = \{\mathbf{c}^{(i)}\}_{i=0}^N$ is the set of configurations a NAS/HPO search method traversed. As a result, the total search energy $e_s$ cost is defined as:
>
> $e_s(A) = \sum_{\mathbf{c}^{(i)} \in A} e_{ep}(\mathbf{c}^{(i)}) * \mathbf{c}^{(i)}_{n}$
>
> where $n$ is the index of $\mathbf{c}$ that stores the number of total training epochs to train the deep model under configuration $\mathbf{c}$. A detailed definition of the search energy cost is added to the appendix of the revised paper.
>
> Furthermore, per the reviewer's suggestion, we elaborate on the details of data collection in the "Details of Data Collection" section in Appendix B.3. Specifically, we provide the code on how to measure the energy cost and the details of the machines used to collect energy consumption related information.
> |Property Name| Value|
> |------------------|------------------------------------------------------|
> | CPU              | Intel(R) Xeon(R) CPU E5-2690 v3   @ 2.60GHz 2600 MHz |
> | Memory-GB        | 112                                                  |
> | Operation system | Linux Ubuntu 20.04 LTS                               |
> | Hard drive-GB    | 1000                                                 |
> | GPU              | $1\times$Nvidia Tesla V100 with 32 GB memory                  |
>
> ### **Number of actually evaluated samples**
> > **“The reviewer could not find the number of actually evaluated samples to build the surrogate model. ”**
>
> We agree with the reviewer that the number of samples actually evaluated is important. In fact, we have reported this number in "Section 2.5 Surrogate Benchmark Evaluation" in the original submission, which says:
> *"The sampled configurations on CIFAR10 and TinyImageNet are split into training, validation, and testing sets containing 11597, 1288, and 1000 samples respectively."*
>
>
> ### **The advantage of Bézier Curves**
> > **“we can use the model that predicts the coefficients of polynomial functions of order  instead of predicting the control points of Bezier curves”**
>
> Compared to general $n$ order polynomial functions, the coefficients of the Bézier Curve are explainable and have real-world semantics (i.e. the control points that define the curvature). As a result, we can leverage the prior knowledge of the learning curve by adding constraints to the control points and fitting a better learning curve. For example, in our implementation, we constrained the starting and ending points of the learning curve to make the accuracy value stay within the $[0, 1]$ range.
>
> Empirically, we conduct an ablation study in which instead of predicting the Bézier Curve control points, directly predicts the coefficients and intercept of polynomial functions. However, we observe that for polynomial functions of higher order (n=4), the model is almost impossible to fit. The possible reason is that the scales of the parameters differ too much, and the magnitude of the coefficients varies widely, making it difficult to learn the model. When we set n to 2, the results are as follows:
>
> | Degree | Avg.R2 | Avg.KT | Final.R2 | Final.KT |
> |--------|--------|--------|----------|----------|
> | n=2    | 0.0437 | 0.547  | -2.66    | 0.182    |
>
> In contrast, regardless of the order of the bezier's curve, the size of the control points is basically on the same order of magnitude and the model can be easily fitted (as shown in Table 7 of Appendix B.4). We add the above into the Appendix B.4.

---

> ### Author Response · Authors · 2022-11-18
> **Response to Reviewer QejS (Part 1/3)**
>
> Thank you very much for your feedback and for recommending acceptance of our paper. Below we try to address all your concerns.
>
> ## **Small "Real" Dataset**
>
> > **"It might be better to provide not only the large-scale surrogate dataset but also the small "real" dataset"**
>
> As introduced in our paper, a sampled real measured dataset containing $13885\times2$ models is collected to expand the entire configuration space with the surrogate model. This sampled dataset may also be used as a small "real" dataset for some HPO and NAS methods such.
>
> Considering the suggestion of the reviewer, we think that the reviewer may wish to provide a small **tabular** benchmark that exhausts the entire space rather than just a subset of sampled points. Therefore, we redefine a very small joint search space with a size of 500. We have added a detailed introduction and analysis of the real small dataset into appendix E. Specific contexts are as follows:
> *"Besides providing a large-scale proxy benchmark and the tens of thousands of sampling points used to construct it, we also provide a small real tabular benchmark. As shown in Table 9, we redefine a very small joint search space with a size of $50\times10=500$. As with the previous tabular benchmark, we evaluate all models within this space."*
>
> Table: Overview of the small tabular search space
>
> |     Type        |     Hyperparameter      |     Range                                                             |     Quantize    |     Space    |
> |-----------------|-------------------------|-----------------------------------------------------------------------|-----------------|--------------|
> |     RegNet      |     depth               |     [6,15]                                                            |     1           |     10       |
> |                 |     w0                  |     [80, 112]                                                         |     8           |     5        |
> |                 |     wa                  |     20                                                                |     -           |     1        |
> |                 |     wm                  |     2.75                                                              |     -           |     1        |
> |                 |     Group_W             |     16                                                                |     -           |     1        |
> |     Total       |                         |                                                                       |                 |     50       |
> |     Optim       |     Base_lr             |     {0.001, 0.003, 0.005, 0.01, 0.03,   0.05, 0.1, 0.3, 0.5, 1.0}    |      -          |     10       |
> |                 |     Max_epoch           |     {100}                                                             |     -           |     1        |
> |                 |     Lr_polic            |     {'cos’}                                                           |     -           |     1        |
> |                 |     Choise of optimizer    |      sgd                                                              |     -           |     1        |
> |     Training    |     augment             |     None                                                              |     -           |     1        |
> |     Total       |                         |                                                                       |                 |     10       |

---

### Author Response · Authors · 2022-11-18
**To all reviewers**

We sincerely appreciate the reviewers' careful reading, constructive questions, and suggestions. We are encouraged that you found our benchmark dataset comprehensive and well-designed (Reviewer bERE), valuable for the community (Reviewer QejS) and be very helpful for saving energy for future research (Reviewer ZDyf and Reviewer uqiJ).
We believe that the EA-HAS-Bench takes one of the most essential steps in developing EA-NAS methods to provide a benchmark that makes EA-NAS research more reproducible and accessible. According to the opinions of peer reviewers, we have made the following major improvements to the paper:

+ provide the training, validation, testing curve, details and codes of the dataset collection and experiments, and the API on how to use EA-HAS-Bench, the total energy consumption to build EA-HAS-Bench.
+ add more empirical evaluations and explanations in Section 5
+ add more analysis and details for energy cost measurements
+ refine the writing to explain all unclear points

We have written point-by-point replies that explain how we addressed the reviewers' technical comments. ***All the major changes are highlighted in red in the revised PDF.*** We again thank the reviewer for recommending the acceptance and for their comments to improve the clarity and reproducibility of the paper.

---

### Decision · Program_Chairs · 2023-01-20

**Decision:**

Accept: notable-top-25%

**Justification For Why Not Higher Score:**

Given the existence of the JAHS-Bench paper, this now "only" adds the energy measurements.

**Justification For Why Not Lower Score:**

The benchmark will be very useful for the NAS communitiy.

**Metareview: Summary, Strengths And Weaknesses:**

This paper introduces a novel joint NAS+HPO benchmark that also includes measurements of energy. All reviewers judged this to be very helpful and gave acceptance scores. Joint NAS + HPO is very important, as also recently addressed in the NeurIPS datasets & benchmarks paper on JAHS-Bench-201. I agree with the reviewers and recommend acceptance.

**Note From Pc:**

if the above contains the word "oral" or "spotlight" please see: "oral" presentation means -> notable-top-5% and "spotlight" means -> notable-top-25%. As stated in our emails, we are disassociating presentation type from AC recommendations